# Asynchronous fate decisions by single cells collectively ensure consistent lineage composition in the mouse blastocyst

Néstor Saiz[1], Kiah M. Williams[1], Venkatraman E. Seshan[2] & Anna-Katerina Hadjantonakis[1]

Intercellular communication is essential to coordinate the behaviour of individual cells during organismal development. The preimplantation mammalian embryo is a paradigm of tissue self-organization and regulative development; however, the cellular basis of these regulative abilities has not been established. Here we use a quantitative image analysis pipeline to undertake a high-resolution, single-cell level analysis of lineage specification in the inner cell mass (ICM) of the mouse blastocyst. We show that a consistent ratio of epiblast and primitive endoderm lineages is achieved through incremental allocation of cells from a common progenitor pool, and that the lineage composition of the ICM is conserved regardless of its size. Furthermore, timed modulation of the FGF-MAPK pathway shows that individual progenitors commit to either fate asynchronously during blastocyst development. These data indicate that such incremental lineage allocation provides the basis for a tissue size control mechanism that ensures the generation of lineages of appropriate size.

[1] Developmental Biology Program, Sloan Kettering Institute, Memorial Sloan Kettering Cancer Center, 1275 York Avenue, New York, New York 10065, USA. [2] Department of Epidemiology and Biostatistics, Memorial Hospital, Memorial Sloan Kettering Cancer Center, New York, New York 10065, USA. Correspondence and requests for materials should be addressed to A.-K.H. (email: hadj@mskcc.org).

Coordinated cell behaviour is an essential characteristic of multicellular organisms. During embryonic development, cellular proliferation, death and differentiation must be precisely coordinated, to generate an organism of the appropriate size and cellular composition. Embryos of different animal taxa display a range of regulative abilities that allow them to produce consistent, reproducible structures, even when faced with changes in cell number or morphological alterations[1]. However, the cellular bases for these regulative abilities are poorly understood.

The preimplantation mammalian embryo is a paradigm of regulative development and self-organization. During pre-implantation development, the fertilized egg gives rise to the blastocyst—the embryonic structure capable of implanting into the uterus—without the need for maternal input. The blastocyst stage is highly conserved across mammals and comprises two extraembryonic epithelia, trophectoderm (TE) and primitive endoderm (PrE, or hypoblast in non-rodents), both of which encapsulate the embryonic lineage: the pluripotent epiblast (EPI). The EPI gives rise to most somatic cell types *in vivo* and to embryonic stem (ES) cells *in vitro*[2–6]. The descendants of the TE and PrE form the embryonic part of the placenta and the yolk sacs, respectively, and they nourish and pattern the EPI as it develops into the fetus. Defects in blastocyst development often result in embryonic lethality. Thus, correct blastocyst formation is critical for successful development.

Cell differentiation during preimplantation development is largely dictated by intercellular interactions and spatial cues, rather than pre-determined, and consequently the embryo is capable of accommodating changes in cell number and other experimental manipulations, while developing to term[7–12]. In the mouse, over the first ∼2.5 days of development (∼E2.5), cleavage of the zygote results in the formation of the 8–16 cell morula. The cells located on the surface of the morula acquire TE identity, whereas cells located on the inside of the morula comprise the inner cell mass (ICM), which gives rise to the PrE and EPI (reviewed in refs 13,14).

Lineage choice in the ICM is partly determined by lineage-specific transcription factors. ICM cells in the early blastocyst (E3.0–3.25) co-express marker genes for both PrE and EPI, such as the zinc-finger GATA-binding family member GATA6 and the homeodomain transcription factor NANOG, respectively[15–17]. GATA6 and NANOG are required for the specification of each lineage[18–23]. However, *Fgf4* (fibroblast growth factor-4) is the first gene to be differentially expressed within the ICM[24,25] and its activation of FGF receptors (FGFRs) on neighbouring cells is thought to lead to mutually exclusive expression of PrE and EPI markers at later blastocyst stages (E3.75–E4.0)[15–17,24,26,27]. Whereas no signal is known to be required for EPI speci-fication, FGF4 is the signal necessary for ICM cells to acquire PrE identity[28–30]. FGF4 activates the receptor tyrosine kinase (RTK)–mitogen-activated protein kinase (MAPK)–extracellular signal-regulated protein kinase (ERK) pathway, thus maintaining GATA6 expression and triggering the PrE-specific genetic programme[21,24,26–34].

The key elements driving the specification of PrE and EPI within the ICM (GATA6, NANOG and FGF4–RTK–ERK) and the sequential phases of gene expression (overlapping and mutually exclusive) have been established. However, studies addressing the functional significance of these two phases and how they affect the regulative nature of the blastocyst have yielded somewhat contradictory results. Lineage tracing and chimera experiments have argued that EPI cells exhibit restricted developmental potential from very early stages of blastocyst development (∼E3.25–E3.5)[26,35], and can only contribute to the EPI lineage when placed into a host embryo. By contrast, pharmacological modulation of the FGF4–RTK pathway led to

the proposal that all ICM cells remain plastic until the late blastocyst stage (∼E4.0) and can differentiate into either PrE or EPI[33]. However, this study did not consider the precise developmental stage of the experimental embryos, making it difficult to associate experimental outcome to develop-mental stage. Furthermore, these studies did not undertake a single-cell resolution analysis of all cells within any given embryo.

In the present study, we have probed the plasticity of ICM cells at single-cell resolution and at defined stages of mouse blastocyst development, taking advantage of our recently developed single-cell image analysis pipeline[22,36–38]. Our data reveal that PrE and EPI are formed through incremental allocation of cells from a common ICM progenitor pool. Timed modulation of the FGF4–RTK–ERK pathway showed that individual ICM cells acquire PrE or EPI fate in an asynchronous manner, and that the outcome of signal modulation can be predicted by the size of the pool of uncommitted ICM cells. Our data support the notion that lineage specification in the ICM is a unidirectional event, and that PrE and EPI cells cannot undergo cell fate switches *in vivo* once specified[37]. Moreover, we show that the lineage composition of the ICM in late blastocysts is highly consistent and thus we hypothesize the existence of a mechanism for tissue size control operating in the mammalian blastocyst. Therefore, we propose that the progressive loss of progenitors, through incremental allocation to the PrE and epiblast lineages, endows the mammalian embryo with a mechanism to ensure the gene-ration of tissues of appropriate size, and provides a cellular basis for regulative development.

## Results

**EPI and PrE emerge incrementally from a progenitor pool**. The current model for lineage specification in the ICM of the mouse blastocyst involves three successive phases based on gene expression patterns and changes in cell position: coexpression of PrE and EPI markers is followed by mutually exclusive expression of markers in cells that are initially scattered in a salt-and-pepper distribution and which subsequently sort into two spatially coherent populations[13,15,24,26,39,40] (Fig. 1a). This view is conceptually simple, but masks the heterogeneity found within the ICM, where cells coexpressing both markers (GATA6 and NANOG) exist alongside cells exhibiting mutually exclusive expression at different developmental stages[15,22–24] (Fig. 1b).

We recently developed an image analysis pipeline to quantify fluorescence intensities of each and every cell within preimplantation embryos in a semi-automated, unbiased manner[22,36–38,41]. Building on this pipeline, we sought to determine the identity of all ICM cells of blastocysts spanning all stages, from 32 cells until implantation, at embryonic day (E)4.5 (150–200 cells; Supplementary Fig. 1a). Blastocysts were fixed on collection, immunolabelled and Z-stacks of confocal images spanning the entire embryo acquired (Fig. 1b and see Methods). All nuclei within each Z-stack were then segmented and fluorescence levels for all channels measured using the MINS algorithm[36].

We processed data for 139 blastocysts, which we grouped into 5 developmental stages, based on total cell number: 32–64 cells (N = 45 embryos), 64–90 cells (N = 41), 90–120 cells (N = 32), 120–150 cells (N = 14) and >150 cells (N = 7; Supplementary Fig. 1a). Embryos at the 32–64-cell stage correspond to early blastocysts, just after initiation of cavitation, whereas embryos over 150 cells correspond to the peri-implantation (E4.5) stage. We tested several approaches to determine the identity of ICM cells in these embryos (see Methods for details). A thresholding approach using fluorescence intensity levels (a readout of nuclear

protein concentration) of GATA6 and NANOG was effective at separating ICM cells into four populations (Fig. 1c): EPI (cells expressing only NANOG), PrE (only GATA6), double positive (DP, both NANOG and GATA6) and double negative (DN, expressing neither of the markers). However, this approach is deterministic and requires manual establishment of thresholds for each experiment on highly spread, linear data. To perform a more unbiased, data-driven cell classification, we applied a *k*-means clustering approach to identify cell populations using log-transformed values of GATA6 and NANOG concentrations

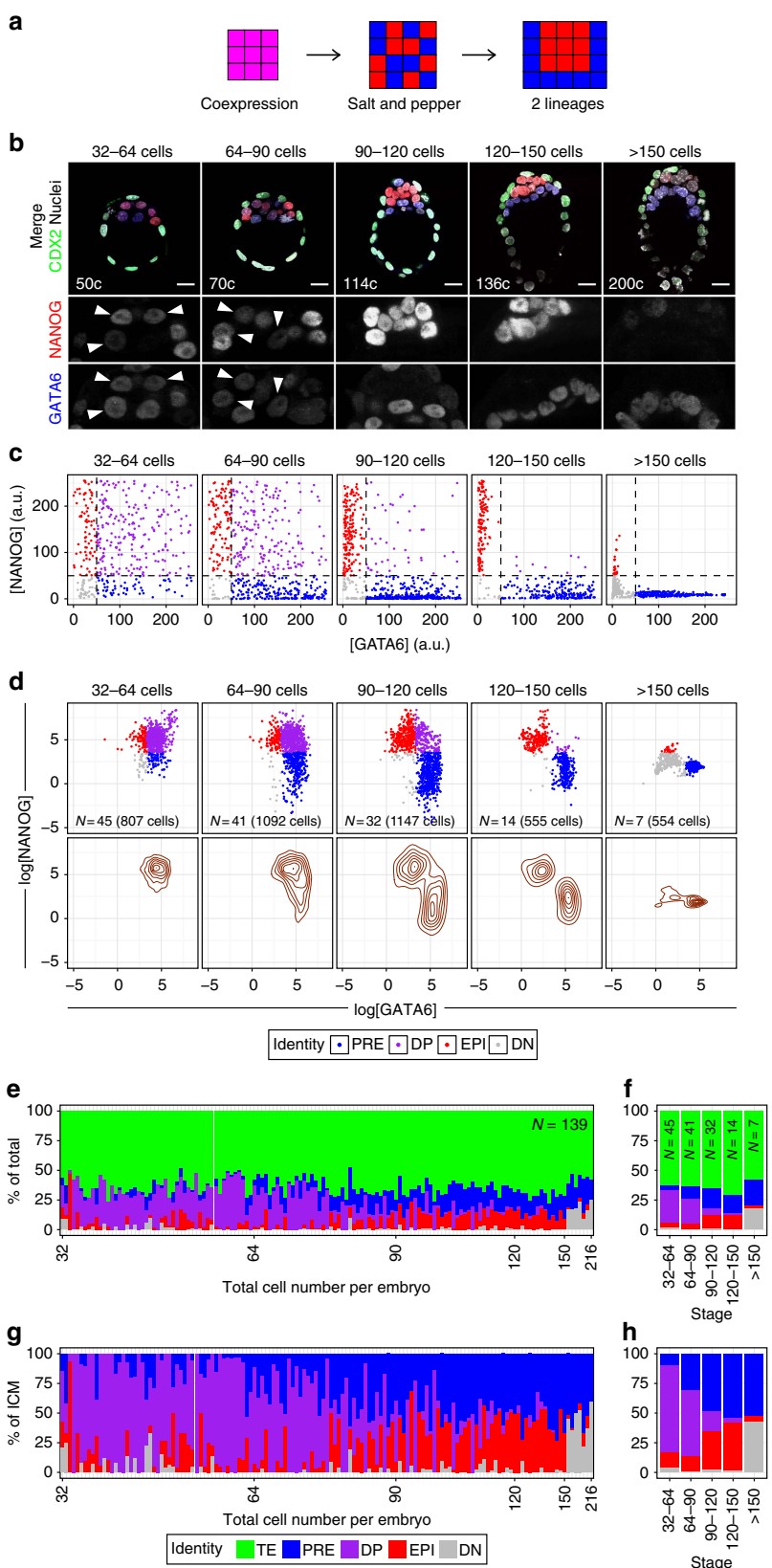

(Fig. 1d). We used the intersection of early (32–64 cells, with high proportion of DP cells) and late (120–150 cells, with clearly distinct PrE and EPI clusters) blastocyst stages, to determine the cluster centres of EPI, PrE, DP and DN populations, and applied these to identify cells across all stages. This approach yielded nearly identical results to the thresholding method, while being adaptable to different experimental setups (see Methods).

Our analysis revealed that cells expressing both GATA6 and NANOG (DP) coexisted within the ICM with PrE and EPI cells in blastocysts of up to 90–120 cells (E3.75–E4.0; Fig. 1b–d). When assessing the composition of the ICM as a function of developmental stage (total cell number), we found a progressive conversion of this DP compartment into PrE and EPI (Fig. 1e–h and Supplementary Fig. 1b,d), suggesting the DP compartment represents a pool of bipotent progenitors for both PrE and EPI. By contrast, the average TE:ICM ratio remains constant across developmental stages (Fig. 1e,f and Supplementary Fig. 1c). Notably, we observed a stabilization of ICM composition at around 60% PrE and 40% EPI in embryos over 100 cells, which displayed no or very few DP cells ($\sim$5%), and where all ICM cells are committed to their respective fates[2,35] (Supplementary Fig. 1e). A reduction in NANOG levels in blastocysts >150 cells (Fig. 1c) leads to an increase in DN cells (GATA6−, NANOG−), at the expense of the EPI compartment (Fig. 1b,d). DN cells are nonetheless located within the ICM and we consider them to be late-stage EPI, as they retain OCT4 expression and *Nanog* is downregulated as the EPI transitions from a naive to a primed pluripotent state[22,32,42].

Our single-cell level analysis thus revealed heterogeneity in ICM composition, where DP cells coexist with GATA6+ (PrE) and NANOG+ (EPI) cells throughout most of blastocyst development. As these DP cells are lost as PrE and EPI arise, we posit the DP compartment represents a transient, progenitor state for both lineages. In such a scenario, only the fate of these progenitors would be affected by changes in FGF4–RTK–MAPK pathway activity, whereas PrE or EPI cells might be insensitive to this signal.

**Single ICM progenitors become PrE or EPI asynchronously.**
We next investigated whether the temporal depletion of ICM progenitors determined the outcome of modulating the FGF4–RTK–MAPK pathway. We subjected blastocysts to a series of treatment regimes (defined by developmental stage and time in culture; Fig. 2a) in different conditions (growth factors or small-molecule inhibitors). We collected individual litters of wild-type embryos at sequential stages of blastocyst development (Fig. 2a), fixed a subset of littermates on collection (for staging purposes)

and cultured the remaining embryos in the presence of either saturating amounts of FGF4 or inhibitors of the RTK–MAPK pathway (MAPK/ERK kinase (MEK) inhibitor (MEKi hereinafter) or FGFR inhibitors (FGFRi))—see Methods for details.

Embryos cultured for 48–72 h in either FGF4 or the MEKi PD0325901 from the eight-cell stage develop an ICM composed entirely of either PrE (FGF4) or EPI (MEKi) cells[32,33]. Furthermore, PD0325901 maintains ground-state pluripotency in ES cells[43]. Using this approach we obtained blastocysts with ICMs composed of either GATA6+ PrE cells only (FGF4) or NANOG+ EPI cells only (MEKi; Fig. 2b-d; '8-cells + 48 h'). The FGFRi AZD4547 (ref. 44) and SU5402 (ref. 45) also cause ERK1/2 inhibition in ES cells[43,46,47] and yield comparable results to MEKi (Fig. 2b–d and Supplementary Fig. 3a–c). We therefore applied these conditions to alter FGF4–RTK–MAPK signalling at sequential stages of blastocyst development.

To establish each treatment regime, we used four of the developmental stages described above (Figs 1b–d and 2a): 32–64 cells (E3.25–E3.5, DP cells $\sim$50–75% of the ICM); 64–90 cells ($\sim$E3.5, DP cells $\sim$25–50% of the ICM); 90–120 cells ($\sim$E3.75, DP cells <25% of the ICM) and >120 cells ($\sim$E4.0), where the amount of DP cells is 0 or negligible ($\sim$5%). We cultured all embryos until they reached 120–170 cells (Fig. 2a and Supplementary Fig. 4a).

For all treatment regimes, control embryos displayed spatially segregated EPI and PrE populations, where NANOG and GATA6 were mostly mutually exclusive (Fig. 2b,c and Supplementary Fig. 3b). We used the clusters found in controls to assign ICM lineages in treated embryos (see Methods). Unexpectedly, many embryos collected at the 32–64-cell stage and cultured for 30 h with FGF4 displayed a small number of NANOG+ EPI cells located on the inside of an ICM otherwise composed of PrE cells (Fig. 2b–d and Supplementary Figs 3d and 4b). Conversely, most embryos cultured with MEKi displayed some GATA6+ PrE cells on the surface of an ICM composed predominantly of EPI cells (Fig. 2b–d and Supplementary Figs 3d and 4b). Most embryos collected at the 64–90-cell stage and treated with FGF4, displayed a significant number of EPI or DN cells within the ICM, whereas embryos treated with MEKi or FGFRi exhibited many PrE cells on the ICM surface (Fig. 2b–d and Supplementary Figs 3a–d and 4b), consistent with previous observations[22,32]. This effect was even more pronounced in embryos collected at 90–120 cells and treated in either condition (Fig. 2b–d and Supplementary Figs 3a–d and 4b). Embryos treated from the 120–150-cell stage were indistinguishable from untreated, control embryos, except for the maintenance of high levels of NANOG on MEKi/FGFRi treatment (Fig. 2b–d and Supplementary Figs 3a–d and 4b). Therefore, the ability of FGF4–MAPK activity to affect the

**Figure 1 | Incremental allocation of cells to PrE and EPI.** (**a**) Current model of PrE and EPI specification, representing changes in expression of GATA6 and NANOG, and spatial distribution of PrE (blue) and EPI (red) cells. (**b**) Representative immunofluorescence images of blastocysts collected at sequential developmental stages. NANOG (EPI) and GATA6 (PrE) shown in grayscale in ICM magnifications for each embryo. In merged image, CDX2 marks the TE lineage. Total number of cells ('c') of the embryo shown is indicated in the merged image. Arrowheads mark DP cells coexpressing NANOG and GATA6. All images are 5 μm Z-projections. (**c**) Scatter plots for fluorescence intensity levels of GATA6 and NANOG in individual ICM cells of embryos collected at sequential stages, as in **b**. Values shown in linear scale after inverting the log-transformed, corrected values for each ICM cell (see Methods). Axes are capped at 255 a.u., as original values. Cell identity (colour coded) was assigned using a thresholding approach based on linear-scale values (see Methods). Thresholds (50 a.u.) are indicated as dashed lines. (**d**) Scatter plots for same data as in **c**, represented as logarithm. Cell identity was assigned using a *k*-means clustering approach (see Methods). Contour lines are provided as density estimators. Number of embryos (*N*) and cells analysed are given on each plot. (**e**) Lineage composition, shown as % of the total, for each embryo analysed in **b–d**, ordered by increasing total cell number. TE and ICM identity was assigned manually. Total number of embryos analysed (*N*) is indicated. (**f**) Average lineage composition, as in **e**, binned in the developmental stages indicated. The number of embryos per bin (*N*) is indicated. (**g**) ICM composition, shown as % of the ICM, for each embryo analyzed in **b–d**, ordered by increasing total cell number. Cell identity assigned as in **d**. (**h**) Average ICM composition, as in **g**, binned in the developmental stages indicated. Colour coding is indicated. DN, double negative (GATA6−, NANOG−); DP, double positive (GATA6+, NANOG+); EPI, epiblast (NANOG+); PRE, primitive endoderm (GATA6+); TE, trophectoderm. For a description of the criteria used to correct fluorescence levels along the Z axis and to determine cell identity, see Methods. Scale bar, 20 μm.

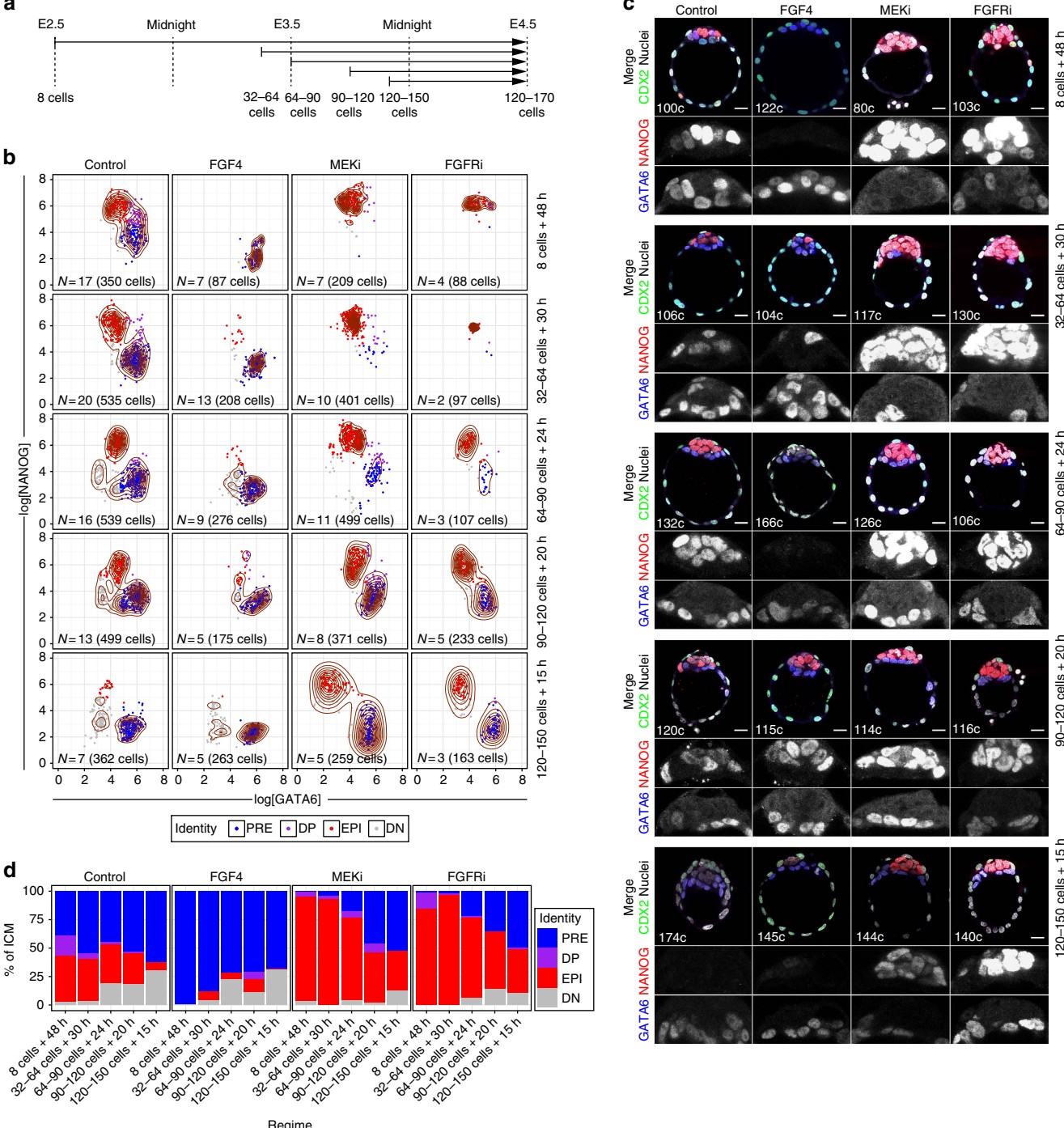

**Figure 2 | ICM cells become committed to PrE or EPI in an asynchronous manner.** (**a**) Treatment regimes embryos were subject to. Arrows represent the length of the culture period (in either condition), from embryo collection (beginning of the arrow) to fixation (arrowhead). Developmental stage at the time of collection and fixation are indicated below the arrows. Developmental stage at the time of collection was determined based on that of fixed littermates (shown within Fig. 1 and see Methods). (**b**) Scatter plots for the levels of GATA6 and NANOG (expressed as logarithm) in individual ICM cells in all embryos at the end of the culture period. Columns represent treatment condition and rows represent treatment regimes as represented in **a**. Control: KSOM or KSOM + 1 μg ml$^{-1}$ of heparin (addition of heparin had no detectable effect on control embryos); FGF4: KSOM + 1 μg ml$^{-1}$ rhFGF4 + 1 μg ml$^{-1}$ of heparin; MEKi: KSOM + 1 μM PD0325901; FGFRi: KSOM + 1 μM AZD4547. Contour lines have been overlaid as density estimators. Cell identity (colour coded) was assigned using a *k*-means clustering approach that used the distribution of cells in Control embryos to establish cluster centres (see Methods). Owing to the data distribution, some cells belonging to the PrE in some Control groups were classified as DN. Number of embryos (*N*) and cells analysed are given on each plot. (**c**) Representative immunofluorescence images of embryos analysed in **b**. NANOG (EPI) and GATA6 (PrE) are shown in grayscale in ICM magnifications. In merged image, CDX2 marks the TE lineage and 'c' indicates the total number of cells of the embryo shown. All images are 5 μm Z-projections. (**d**) Average ICM composition at the end of the culture period for embryos treated in each of the conditions (Control, FGF4, MEKi and FGFRi) and regime (indicated on the X axis), shown as % of the ICM. Colour coding is indicated. DN, double negative (GATA6 − , NANOG − ); DP, double positive (GATA6 + , NANOG + ); EPI, epiblast (NANOG + ); PRE, primitive endoderm (GATA6 + ). For a description of the criteria used to correct fluorescence levels along the Z axis and to determine cell identity, see Methods. Scale bar, 20 μm.

composition of the ICM is progressively lost over developmental time. Interestingly, the proportion of single-positive cells found at the beginning of treatment was a good predictor of treatment outcome, barring potential variation due to differential proliferation rates and/or inter-embryo variability (Figs 1h and 2d). These data lead us to conclude that the differential response to FGF modulation that we observe is a result of the developmental stage of the embryo at the time of treatment (that is, the amount of progenitors present in the ICM).

Notably, NANOG levels were affected by all treatment conditions, irrespective of the treatment regime: FGF4 treatment reduced NANOG levels in EPI cells—thus increasing the % of DN cells (Fig. 2b,d and Supplementary Fig. 5a)—whereas MEKi or FGFRis maintained elevated levels of NANOG (Fig. 2b,c and

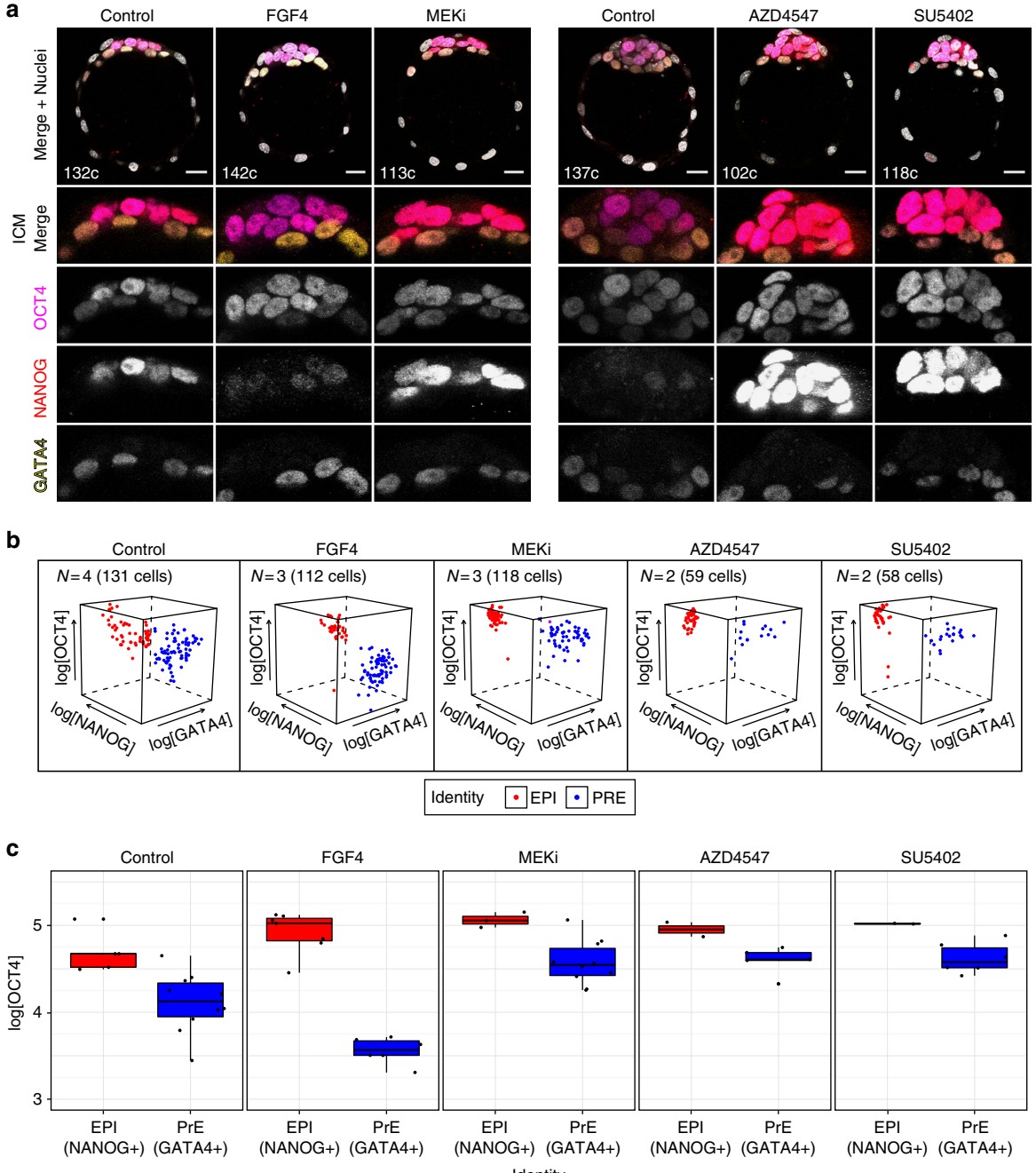

**Figure 3 | ICM lineages in treated blastocysts express markers of *bona fide* PrE and EPI.** (**a**) Representative immunofluorescence images of blastocysts after treatment from the late (90–120 cells; FGF4 and MEKi) or mid (64–90 cells; AZD4547 and SU5402) blastocyst stage as indicated in Fig. 2a. OCT4 (ICM), NANOG (EPI) and GATA4 (PrE) are shown in grayscale in ICM magnifications. ICM and TE lineages were determined manually based on OCT4 expression and cell position. Total number of cells ('c') for the embryo shown is indicated in the merged image. All images are 5 μm Z projections. (**b**) Scatter plots for the levels of OCT4, NANOG and GATA4 (as logarithm) in individual ICM cells in all embryos treated as indicated. PrE and EPI identities were assigned manually. ICM cells cluster based on GATA4 and NANOG/OCT4 levels. (**c**) Boxplots showing OCT4 levels (as logarithm) for embryos shown in **a**, grouped by treatment condition. Each dot represents the average log[OCT4] for all cells in the corresponding lineage (EPI or PrE) per embryo. Colour coding is indicated. EPI, epiblast (NANOG + ); PRE, primitive endoderm (GATA4 + ). For a description of the criteria used to correct fluorescence levels along the Z axis, see Methods. Scale bar, 20 μm.

Supplementary Figs 3a,b and 5a). Therefore, EPI cells (NANOG +) respond to activation of the pathway not by differentiating to PrE, but progressing from a naive (high expression of NANOG) to a primed EPI state (low NANOG)[32,48].

Taken together, our data indicate that FGF modulation can only alter the fate of ICM progenitors (DP cells) and not EPI or PrE cells. They suggest that single-positive cells (NANOG + or GATA6 +) may be committed and support previous findings showing that EPI cells, once specified, do not change fate *in vivo* in the blastocyst[37].

**PrE and EPI maintain lineage markers on FGF modulation**. To assess whether modulation of the FGF–RTK–MAPK pathway altered marker expression beyond GATA6 and NANOG, we treated mid to late blastocysts (80–100 cells) as described above and assessed the expression of GATA4 and OCT4, which are later PrE and ICM/EPI markers, respectively[15,16,49,50]. Embryos treated in all conditions expressed GATA4 in the PrE, NANOG in the EPI and OCT4 throughout the ICM (Fig. 3a). Although OCT4 labelled specifically all ICM cells (Fig. 3a and Supplementary Fig. 5b), its levels were higher in EPI than in PrE cells in all conditions, as shown for peri-implantation blastocysts (Fig. 3b,c)[22,35]. Notably, treatment with FGF4 caused a specific decrease in OCT4 levels in PrE cells (Fig. 3c and Supplementary Fig. 5b), suggesting FGF4 may be involved in its downregulation in the PrE during normal development. Conversely, inhibition of MEK or FGFRs increased the levels of OCT4 in PrE cells when compared with control conditions (Fig. 3c and Supplementary Fig. 5b), further indicating that activity of this signalling pathway can play a role in fine-tuning gene expression in PrE and EPI cells, without affecting their identity.

Overall, these data further indicate that ICM cells that acquire PrE or EPI fate do not switch fate as a result of changes in FGF–RTK–MAPK activity, as they maintain the marker expression pattern of *bona fide* PrE and EPI cells.

**Mouse blastocysts achieve a consistent ICM composition**. We subjected embryos to a range of culture regimes, where they were collected at sequential stages between E2.5 and E4.0, and kept in culture for different time periods (Fig. 2a). However, the composition of the ICM in control embryos at the end of the culture period was consistent across culture regimes (Figs 2d and 4a). Although the ICM size decreased with long culture times (Fig. 4b,c), the average 3:2 ratio of PrE to EPI cells was preserved irrespective of culture time (55–60% PrE to 45–40% EPI cells; Fig. 4a and Supplementary Fig. 6b). Minor deviation from this ratio was likely to be due to *in vitro* culture-induced developmental delay; for instance, control embryos with <100 cells maintained a small proportion of progenitor (DP) cells (Supplementary Fig. 6b), whereas those more developmentally advanced presented a higher proportion of primed, NANOG − EPI cells (DN) (Fig. 4a,c and Supplementary Fig. 4b), as did ∼E4.5 blastocysts (Fig. 1h and Supplementary Fig. 1b). These data therefore suggest that the size of each compartment scales with that of the ICM to achieve a consistent ICM composition.

To test this hypothesis, we experimentally manipulated the size of the ICM (Fig. 4d). Mouse embryos up until the eight-cell stage can accommodate changes in cell number, as well as dissociation and rearrangement, while retaining the ability to develop into blastocysts and implant into the uterus[7,8,10,51–55]. To reduce the size of the embryo (and thus that of the ICM), we collected embryos at the eight-cell stage (E2.5), dissociated blastomeres and re-aggregated them into quartets of four ('half embryos') or groups of eight blastomeres, as controls ('single embryos';

Fig. 4d). Conversely, to increase embryo size, we aggregated pairs of eight-cell stage embryos ('double embryos'; Fig. 4d)[51–53].

After 48 h in culture, all embryos developed into blastocysts (Fig. 4e) of correspondingly half or double the size of control embryos (Fig. 4f) as previously reported[52,53]. Accordingly, the TE and ICM of these embryos were also half or double the size than those of control embryos, respectively (Fig. 4g,h). However, despite the halving or doubling of ICM size, both half and double embryos had a proportion of PrE and EPI cells comparable to that of control embryos grown for 48 h from the eight-cell stage (Figs 4i and 2d, and Supplementary Fig. 6a,c) and to that of freshly collected late blastocysts (>120 cells; Fig. 1h). These results show that ICM composition in the mouse blastocyst is consistent, and that the size of each tissue is scalable. The maintenance of a constant PrE to EPI ratio in the ICM of mature blastocysts indicates this proportion might be important for successful development and suggests the presence of an endogenous mechanism to regulate lineage size in the blastocyst.

**EPI cells trigger PrE specification but not vice versa**. Having established that allocation to the EPI and PrE occurs incrementally, we next wanted to determine how this process is initiated. The current view posits that FGF4, secreted by EPI cells, stimulates the RTK–MAPK pathway to maintain GATA6 in neighbouring ICM cells, which acquire PrE fate[17,21–24,26,29,30,33]. However, no growth factor signal has been described that can induce EPI identity. Thus, in the absence of PrE-inducing signals or transcription factors, all ICM cells default to an EPI-like state[22–24,26,29,30]. Although the PrE is necessary for EPI maturation[6,56], these findings suggest PrE cells are not needed for EPI specification, thus predicting that EPI cells should be sufficient to induce PrE differentiation, but not vice versa[23].

To experimentally test this model, we prevented specification of EPI cells by exposing 8-cell stage embryos to FGF4 for 30 h (50–64-cell stage; Fig. 5a). At this stage, all ICM cells displayed GATA6 expression, but had completely lost, or exhibited highly reduced levels of, NANOG (Fig. 5b,c)—unlike culture for 24 h only (Supplementary Fig. 7a–c,d). When embryos were released from FGF4 stimulation and cultured for a further 18 h in control medium, the large majority displayed only GATA6 + PrE cells (Fig. 5d–f and Supplementary Fig. 7e), suggesting that, indeed, PrE cells cannot specify EPI fate among their neighbours. Alternatively, FGF4 may remain bound to heparan sulfate proteoglycans[57] and continue to stimulate the receptors. Embryos transferred from FGF4 to MEKi-containing medium for 18 h contained both EPI and PrE cells (Fig. 5g,h), suggesting that residual extracellular FGF4 may continue to stimulate the RTK–MAPK axis in progenitor cells or else that basal MEK activity is sufficient to prevent NANOG upregulation in these cells.

To prevent PrE specification in early-to-mid blastocysts, we cultured embryos with MEKi from the eight-cell stage for 30 h (Fig. 5a and Supplementary Fig. 7c). Fixed littermates displayed pan-ICM expression of NANOG, although ∼50% of ICM cells showed some level of GATA6 expression (DP cells; Fig. 5b,c,f). However, unlike embryos treated with FGF4, when embryos were released from MEK inhibition after 30 h (∼60 cells; Supplementary Fig. 7c), they were able to regenerate a GATA6 + PrE layer, while maintaining a NANOG + EPI (Fig. 5d–f), presumably due to the presence of DP at the time of release from the inhibitor (Fig. 5f).

These data indicate that EPI cells can induce PrE fate among their neighbours, presumably through activation of the RTK–MAPK pathway by FGF4, whereas PrE cells are unable to non-cell autonomously induce EPI fate. These findings support a model in which EPI cells are the source of FGF4 within the ICM

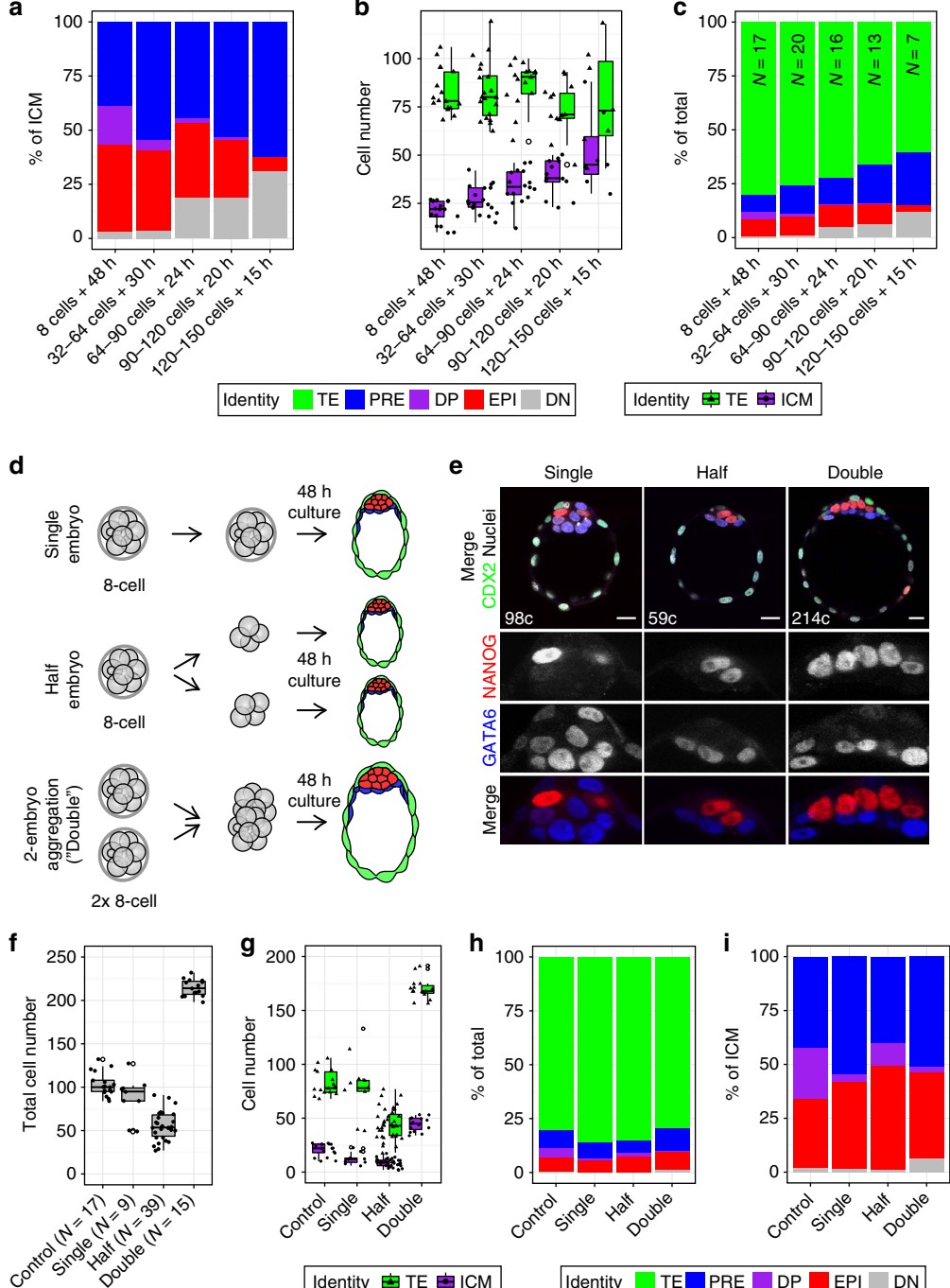

**Figure 4 | Mouse blastocysts achieve a consistent ICM composition.** (**a**) Average ICM composition in embryos collected at sequential stages and cultured in control conditions for the time period indicated, as in Fig. 2d. (**b**) Boxplots showing the number of TE and ICM cells in control embryos shown in **a** and Fig. 2. Each dot represents the number of TE or ICM cells for one embryo. (**c**) Average lineage composition for embryos in **a,b**, and Fig. 2, as % of the total. (**d**) Diagram depicting generation of blastocysts with half ('half embryos') and double the number of cells ('double embryos') using eight-cell stage embryos. Single embryos were eight-cell aggregates for 'half' embryo experiments and intact, eight-cell stage embryos for 'double' embryo experiments (**e**) Representative immunofluorescence images of single, half and double embryos after 48 h in culture. NANOG (EPI) and GATA6 (PrE) shown in grayscale in ICM magnifications and colour coded in merged panel. In top, merged image, CDX2 marks the TE lineage and 'c' indicates the total number of cells of the embryo shown. All images are 5 μm Z projections. (**f**) Box plots showing the total cell number of single, half and double embryos after 48 h in culture, as in **e**. 'Control' embryos correspond to those shown in **a**–**c** and Fig. 2, and are shown for comparison. Number of embryos analysed (N) is indicated. (**g**) Boxplots showing the number of TE and ICM cells in embryos shown in **e,f**. Each dot represents the number of TE or ICM cells for one embryo. (**h**) Average lineage composition for embryos shown in **e,f**, as % of the total. TE and ICM identity was assigned manually. ICM identity was assigned using the same clusters as in Fig. 2. (**i**) Average ICM composition for embryos shown in **e,f**, as % of the ICM. Colour coding is indicated. In boxplots, open circles represent values outside 1.5 × the interquartile range (IQR). DP, double positive (GATA6 + , NANOG + ); DN, double negative (GATA6 − , NANOG − ); EPI, epiblast (NANOG + ); ICM, inner cell mass; TE, trophectoderm; PRE, primitive endoderm (GATA6 + ). For a description of the criteria used to correct fluorescence levels and to determine cell identity, see Methods. Scale bar, 20 μm.

and induce PrE specification. This model suggests that an initial number of EPI cells might be initially specified and trigger lineage divergence within the ICM through activation of the RTK–MAPK pathway among neighbouring progenitors.

## Discussion

Timely and coordinated lineage specification during blastocyst formation is critical for the establishment of the pluripotent EPI and successful development. In this study, we have focused on the

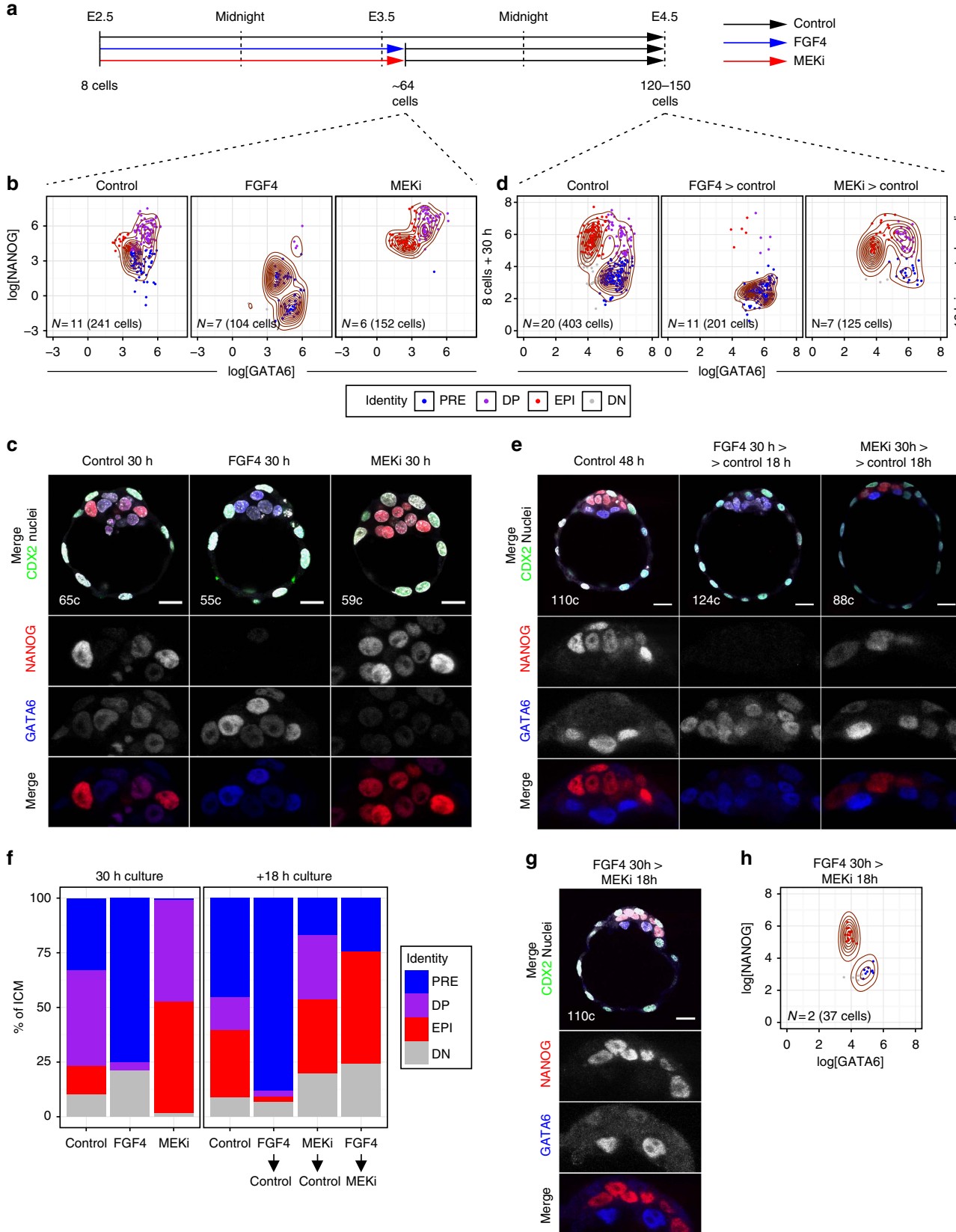

dynamics of EPI (red; Fig. 1a,b) and PrE (blue; Fig. 1a,b) formation within the ICM of the mouse blastocyst. We used our recently developed single-cell resolution image analysis pipeline[22,36], to explore temporal changes in the ability of individual ICM cells to respond to modulation of the activity of the FGF4–RTK–MAPK signalling pathway and to reveal how the behaviours of individual cells collectively drive lineage specification in the blastocyst.

We show that bipotential ICM progenitors (DP) become allocated to EPI or PrE in an asynchronous manner during blastocyst development (Figs 1a,b and 6a). This is further supported by the gradual reduction in the ability of FGF signalling to affect lineage specification as development progresses, defining a window of responsiveness to FGF (from ~32 to 100 cells). Furthermore, our data revealed that ICM composition becomes stabilized in late blastocysts and is conserved regardless of ICM size (Fig. 6a,b). We hypothesize this incremental allocation creates a temporal window of opportunity for the embryo to control ICM composition and scale the size of each lineage with changes in ICM size (Fig. 6a). A corollary of these results is that relative and not absolute lineage size in the blastocyst is critical for successful development, and that a tissue size control mechanism operates to ensure the balanced generation of both cell types. These results support previous studies showing changes in absolute size can be compensated for after postimplantation[52,53], while hinting that relative tissue size is controlled at preimplantation stages.

Activation of the RTK–MAPK pathway by FGF4 is essential for PrE specification and EPI maturation[24,26,27,29,30,32,33]. It has been proposed that an initial phase of EPI and PrE marker coexpression is followed by stochastic cell fate choice, reflected by mutually exclusive expression of NANOG and GATA6 (refs 15,26), where FGF4 would act to induce the expression of downstream PrE markers[21]. In addition, *Fgf4* mutants fail to maintain GATA6 after the early blastocyst stage, giving rise to an ICM with only NANOG+ cells[24,29,30]. Moreover, *Gata6*-null mutant embryos lack the PrE lineage entirely, which cannot be rescued by exogenous FGF4 (refs 22,23). These findings indicate that FGF4 is necessary to activate the PrE programme by maintaining GATA6 and concomitantly downregulating NANOG[22].

Extending this model, the data presented here reveal that this process occurs asynchronously at the single-cell level throughout blastocyst development. Individual ICM cells acquire PrE or EPI fate at different points between the 32- and 100–120-cell stages, and are subsequently unable to change fate despite changes in FGF4–RTK–MAPK pathway activity. Similar observations had been reported previously[22,32], although an explanation for this behaviour had not been proposed. Although we do not elucidate the mechanism underlying cell commitment, this transition

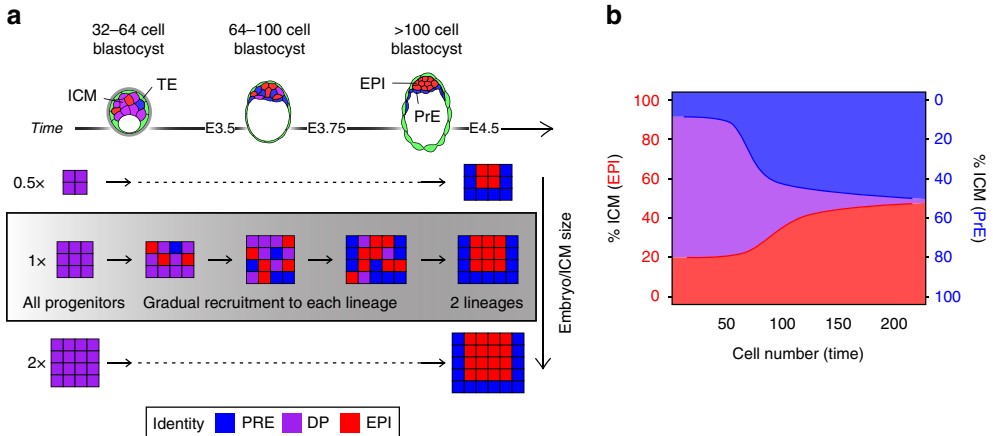

**Figure 6 | Working model.** (**a**) Diagram representing our model of incremental allocation of cells to the PrE (blue) and EPI (red) compartment throughout blastocyst development from a common pool of progenitors (purple). Lineage allocation and cell sorting take place concomitantly. Experimental changes in absolute embryo or ICM size (0.5 × or 2 ×) do not alter the relative proportions of each compartment in the late blastocyst. Cartoons of sequential blastocyst stages are provided as reference. Colour coding is indicated. DP, double positive; EPI, epiblast; ICM, inner cell mass; PrE, primitive endoderm; TE, trophectoderm. (**b**) Diagrammatic representation of temporal changes in ICM composition over time as a result of the asynchronous allocation of progenitors to each the PrE and EPI compartments.

**Figure 5 | EPI cells trigger PrE specification but not *vice versa*.** (**a**) Diagram of the treatment regimes embryos were subject to. (**b**) Scatter plots for GATA6 and NANOG levels (expressed as logarithm) in individual ICM cells, for each treatment condition after 30 h in culture. Contour lines have been overlaid as density estimators. Control: KSOM or KSOM+1 μg ml$^{-1}$ of heparin; FGF4: KSOM+1 μg ml$^{-1}$ rhFGF4+1 μg ml$^{-1}$ of heparin; MEKi: KSOM+1 μM PD0325901. Cell identity (colour coded) assigned using the same clusters as in Fig. 1d. Number of embryos (*N*) and cells analysed are indicated in each plot. (**c**) Representative immunofluorescence images of embryos analysed in **b**. NANOG (EPI) and GATA6 (PrE) shown in grayscale in ICM magnifications and colour coded in the merged panel. In merged image, CDX2 marks the TE lineage and 'c' indicates the total number of cells of the embryo shown. All images are 5 μm *Z* projections. (**d**) Scatter plots for GATA6 and NANOG levels (as logarithm) in individual ICM cells, after release from FGF4/MEKi and culture in control medium for another 18 h. Contour lines have been overlaid as density estimators. Cell identity (colour coded) was assigned using the same clusters used as in Fig. 2b. Number of embryos (*N*) and cells analysed are indicated in each plot. (**e**) Representative immunofluorescence images of embryos analysed in **d**. (**f**) Average ICM composition at the end of the culture period for embryos treated in each of the conditions indicated. (**g**) Representative immunofluorescence images of an embryo treated for 30 h with FGF4 and cultured in MEKi for further 18 h, showing both EPI and PrE layers. (**h**) Scatter plot for GATA6 and NANOG levels (as logarithm) in individual ICM cells for embryos cultured as in **g**. Contour lines have been overlaid as density estimators. Cell identity was assigned as in **d**. Colour coding is indicated. DN, double negative (GATA6−, NANOG−); DP, double positive (GATA6+, NANOG+); EPI: epiblast (NANOG+); PRE, primitive endoderm (GATA6+). For a description of the criteria used to correct fluorescence levels and to determine cell identity, see Methods. Scale bar, 20 μm.

probably involves stabilization of gene expression and the silencing of loci regulating alternative fates[58,59]. Thus, we propose that only GATA6+, NANOG+ (DP) ICM progenitors respond to changes in the RTK–MAPK pathway, and that, in vivo, once committed, PrE and EPI cells can no longer be affected by changes in extracellular FGF4 produced by EPI cells. In this way, lineage committed cells become sheltered from differentiation signals, which could otherwise alter their identity. Notably, the effect of other signalling pathways, which could potentially alter cell fate, has not been addressed here and cannot be ruled out.

We have defined the population of ICM progenitors as a bipotent transition state coexpressing GATA6 and NANOG (DP cells), based on previous observations that the relative ratio of GATA6 to NANOG affects the rate of PrE specification[22], and that the response to modulation of the FGF–RTK pathway in the blastocyst can be predicted by the size of the progenitor pool (this study). However, a definitive marker for this population, if there is one, has not been described and, therefore, the size of this progenitor pool may differ from that of the DP compartment described here. Alternatively, the progenitor state might be dictated probabilistically within the state-space defined by the expression of fate determinants (including, but not restricted to NANOG and GATA6). The type of analysis presented here does not allow for the precise identification and tracking of progenitors over time. Nevertheless, we believe the precise size of this progenitor population does not affect the overall process and developmental framework we propose.

The incremental allocation of cells to each lineage creates a system with the inherent flexibility to respond to exogenous perturbations, providing robustness and a cellular basis to frame the regulative abilities of the preimplantation mammalian embryo. Allocation of cells from a common progenitor pool permits shifting the rate of specification towards PrE or EPI as a function of the population composition at any time (Fig. 6b). This would argue for the existence of an underlying tissue size-control mechanism, explaining the remarkably consistent ICM composition observed in late blastocysts (100–150 cells), when PrE versus EPI lineage specification is complete (this study and ref. 60). Late blastocysts, either fixed on collection (Fig. 1h), cultured for different periods (15–48 h; Fig. 2d) or manipulated to alter ICM size (Fig. 4i), display a consistent ratio of PrE to EPI of around 55–60% PrE to 45–40% EPI (Supplementary Figs 1e and 6b,c). We propose that this ratio has physiological significance, and that controlling it is more important than regulating absolute embryo or lineage size, which can be adjusted after implantation[52,53]. Deviation from this ratio in embryos mutant for PrE factors[22,61] or in embryos with an experimentally reduced EPI[55] indicates a certain tolerance for variation, which might be compensated by a subsequent increase in proliferation postimplantation.

Our data suggest that EPI cells induce PrE specification through the secretion of FGF4, and that PrE cells cannot induce EPI fate, which relies on active inhibition of ERK signalling by the MEK inhibitor (Fig. 5d–h). However, it has been shown that PrE cells are necessary for EPI maturation[6]. Perhaps a subpopulation of EPI cells might need to be specified early on in development, by an as-yet unknown mechanism, to function as a trigger to ICM lineage divergence. However, our data did not allow us to determine whether a fraction of PrE founder cells are also specified independently of FGF4, alongside these proposed EPI founders. Furthermore, these data present limitations, given that embryos treated with FGF4 for 30 h only present insignificant ICM progenitors (DP cells), which would be the substrate for the emergence of EPI cells on release from FGF4. Culture for 24 h only in the presence of FGF4 did not completely abolish the DP population and perhaps, consequently, EPI cells were found in

littermates after further culture in control media (Supplementary Fig. 7d). Live imaging of both populations will reveal the dynamics of this process, the behaviour of ICM progenitors and the presence of the proposed founder cells for each lineage. Whether these hypothetical founder cells arise as the result of stochastic changes in gene expression[15,62], asynchronicity in cell cycle[25,63,64] or some other mechanism (or any combination of them[65]) cannot currently be ascertained.

Previous studies have explored changes in cell plasticity during blastocyst development, with disparate conclusions[26,33,35]. Chimera and lineage-tracing experiments showed that ICM cells contribute to either one of the PrE or EPI lineages, and thus concluded that ICM cells are committed from very early stages[26]. By contrast, subsequent studies involving chimeras and FGF signalling modulation experiments concluded that ICM cells remain plastic until very late blastocyst stages[33,35], although EPI cells were shown to lose plasticity at early blastocyst stages, before PrE cells[35]. Our observations reconcile these previous findings, whereby uncommitted (DP) and committed (EPI or PrE) cells coexist throughout blastocyst development (Fig. 6a,b). These populations cannot, however, be distinguished using single-lineage live-imaging reporters[26,35]. Importantly, in those chimera experiments, exposure of grafted cells to the host environment might alter their behaviour in ways that the experiments presented here cannot account for. The differences between the present and previously published results[33] could arise from differences in embryo staging. In a previous study, embryos were cultured before treatments[33], which causes developmental delay. In the present study, all embryos were collected immediately before treatment and individual litters were staged using reference littermates fixed at the time of collection or at the time of medium exchange. Given the temporal variation in ICM composition, accurate staging of embryos before treatment is critical for the correct interpretation of these results.

Heterogeneity in ICM composition reflects the presence of three dynamic and functionally distinct subpopulations, and suggests a mechanism by which individual cell behaviours are coordinated to achieve a consistent outcome at the population level. Our results lead us to hypothesize the existence of a tissue size-control mechanism operating in the mammalian blastocyst, which allows for control of lineage composition in a self-organizing system and provides a cellular basis to the regulative abilities of the early mammalian embryo.

## Methods

**Mouse husbandry and embryo collection and manipulation.** All animal work was approved by Memorial Sloan Kettering Cancer Center's Institutional Animal Care and Use Committee. Animals were housed in a specific pathogen-free facility under a 12 h light cycle. All embryos used for this study were obtained from natural matings of outbred CD1 virgin females (Charles River) of 4–8 weeks of age, to CD1 studs. Embryos were flushed out of the oviducts (morula stage) or uterine horns (blastocyst stage) using 500 μl of flushing and holding medium (FHM, Millipore)[66]. Zona pellucidae were removed by brief incubation in acidic Tyrode's solution (Sigma)[66]. All live embryo medium and solutions were preheated at 37 °C. Embryos were fixed for 10 min at room temperature in a 4% solution of paraformaldehyde (PFA, BioRad) in PBS. After fixation, embryos were preserved in PBS at 4 °C. Embryos immunolabelled for GATA4 (see below for details on antibody) were fixed overnight at 4 °C in 4% PFA solution and subsequently stored in PBS at 4 °C.

**Embryo culture.** Morula- and blastocyst-stage embryos were cultured on 35 mm Petri dishes (Falcon), within 10–15 μl microdrops of Potassium Simplex Optimized Medium (KSOM-AA, Millipore) at 37 °C in a humidified 5% $CO_2$ atmosphere. Before embryo culture, KSOM medium was buffered in the incubator for at least 15 min. Embryos were cultured in groups of five to ten embryos per drop. Zona pellucidae were removed from blastocysts on collection and before in vitro culture. Morula-stage embryos were cultured within zonae, which were only removed on cavitation (after 24–30 h).

**FGF signalling modulation.** For FGF4–RTK–MAPK pathway stimulation experiments, recombinant human FGF4 (rhFGF4, R&D Systems) was diluted in culture

medium (KSOM) at saturating concentration (1 µg ml⁻¹)[33]. FGF4 solutions and control medium were supplemented with 1 µg ml⁻¹ of Heparin (Sigma). For FGF4–RTK–MAPK pathway inhibition experiments, the MEKi inhibitor PD0325901 (Stemgent) was diluted in KSOM at 1 µM[32], the FGFR1–3 inhibitor AZD4547 (Santa Cruz)[44] was diluted at 1 µM and the pan-FGFRi SU5402 (R&D Systems)[45] was diluted at 20 µM. Control medium for MEK and FGFR inhibition (MEKi and FGFRi, respectively) experiments was KSOM. No differences were observed between embryos cultured in KSOM and KSOM supplemented with 1 µg ml⁻¹ of Heparin.

These treatment conditions (FGF4, MEKi and FGFRi) were applied in several regimes, defined by the developmental stage of the embryo at the time of collection/ start of the treatment period and the length of the culture period (Figs 2a and 5a, and Supplementary Table 1). The developmental stages used were determined based on previous studies[15] and the changes we observed in gene expression and ICM composition in this study (Fig. 1h).

To ensure accurate staging of embryos and to reduce variability due to stage differences between litters of embryos, each modulation experiment was performed on a single litter. Individual litters were collected from single females, two to three embryos per litter were fixed on collection, as reference specimens to assess the developmental stage of the entire litter, and the rest of embryos distributed between control and either FGF4- or MEKi/FGFRi-supplemented medium for culture. All reference littermates are included in the data set shown (Fig. 1 and Supplementary Fig. 1). The treatment regime applied to each experiment (that is, litter) was determined retrospectively based on the average total cell count of reference littermates (Supplementary Table 1). Although the stage of most litters corresponded to that expected at the time of collection, certain experiments were retrospectively re-assigned to a different regime on staging of littermates. As a consequence, the culture period for those litters was either shorter or longer than would have corresponded to their regime. However, we observed that it is the stage at which the treatment begins, not the exact time in culture, that determines the composition of the ICM at the end of the treatment period. Nonetheless, information on both treatment regime (defined by stage of collection) and the length of the period in culture (treatment length) is available for each embryo and litter in the raw data (see below).

To ensure that the differential response obtained for each regime was not due to the progressively shorter periods of exposure to FGF4 or inhibitors, we cultured embryos collected from each stage for longer periods. We observed no difference in the response to treatment as a result (not shown). Moreover, we observed that culture of embryos from the eight-cell stage for only 30 h was sufficient to alter gene expression (Fig. 5b,c) and thus, presumably, lineage identity, comparable to the 48 h cultures shown in Fig. 2 and previously reported[32,33].

**Generation of half and double embryos.** Eight-cell stage embryos were collected on embryonic day E2.5 from CD1 females as described above. To generate half embryos, embryos were denuded by brief incubation in acidic Tyrode's, placed back in FHM, and physically dissociated by forcing them through a finely drawn glass Pasteur pipette. Blastomeres were transferred to microdrops of buffered KSOM-AA and re-aggregated in groups of four cells. Dissociation can cause cell lysis, in which case dead blastomeres were replaced with blastomeres from another embryo. Control embryos for these experiments were made by re-aggregating eight blastomeres after dissociation. It should be noted that some re-aggregated, 'single' control embryos were observed to have fewer cells than intact controls after 48 h of culture (Fig. 4f), which could be due to cell death after reaggregation caused by experimental manipulation. Cell aggregates were cultured in KSOM-AA micro-drops, in depressions made on 35 mm Petri dishes (Falcon) with a DN-09 aggregation needle (BLS, Hungary)[66].

To generate double embryos, eight-cell stage embryos were denuded and placed in pairs in depressions on 35 mm Petri dishes, within KSOM-AA microdrops, the same way as half embryos (Fig. 4d). Control embryos for these experiments were intact, denuded eight-cell stage embryos.

Both half and double embryos were fixed for 10 min in 4% PFA after 48 h in culture and processed as described below.

**Immunofluorescence.** Whole-mount immunofluorescence was performed as described in ref. 67. Fixed embryos were washed (5 min) in 0.1% Triton X-100 (Sigma) in PBS (PBX), permeabilized (5 min) in a solution of 0.5% Triton X-100 and 100 µM glycine (Sigma) in PBS, washed (5 min) in PBX and incubated for 30 min to 1 h at room temperature in 2% horse serum (Sigma) in PBS, as a blocking solution, before primary antibody application, incubated overnight at 4 °C in primary antibodies diluted in blocking solution, then washed three times (3 × 5 min) in PBX before blocking (30 min to 1 h) and incubation in secondary antibodies (1 h). Secondary antibodies used were AlexaFluor (LifeTechnologies) diluted 1:500 in blocking solution. Primary antibodies against GATA6 (goat, R&D Systems, AF1700), GATA4 (goat, Santa Cruz, sc-1237) and OCT4 (mouse, Santa Cruz, sc-5279) were diluted 1:100; anti-CDX2 (mouse, Biogenex, MU392A-UC) was diluted 1:200 and anti-NANOG (rabbit, ReproCell RCAB0002P-F) was diluted 1:500. After washing twice (2 × 5 min) in PBS, DNA was stained with 5 µg ml⁻¹ of Hoechst 33342 (Invitrogen) in PBS.

**Image acquisition and segmentation.** To determine the total cell count of each embryo, Z-stacks of whole embryos were imaged using a Zeiss LSM510 laser-scanning confocal microscope equipped with a Zeiss EC Plan-Neofluar 40 × /1.3 numerical aperture objective with a 0.17 mm working distance. Z-sections were acquired every 1 µm. Semi-automated nuclear segmentation for cell counting and fluorescence intensity measurement was performed using the MATLAB-based algorithm MINS[36], as described[38]. MINS is freely available at http://katlab-tools.org (requires MATLAB license).

The same imaging parameters were used across experiments, as discussed[38] and whenever possible, whole litters were imaged in the same session. Embryos in Figs 1 and 5b,c and Supplementary Fig. 7a,b were imaged using the same parameters. Embryos in Figs 2, 4 and 5d,e, and Supplementary Fig. 3 were imaged using the same parameters, which only differed slightly from the previous cohort in the gain settings for the 561 nm and the 633 nm lasers. Gain and laser power for the 405 nm laser was adjusted as necessary, to capture bright, sharp nuclei, as this facilitates nuclear segmentation, but were not used for any data transformation.

**Data processing and statistical analysis.** Over- and under-segmentation of nuclei by MINS was manually corrected as previously described[38]. Fluorescence values for under-segmented cells (two or more nuclei in one segmentation volume or unsegmented nuclei) were replaced by values that were either copied from an equivalent neighbouring nucleus or directly measured using ImageJ (NIH)[38]. In over-segmentation events (one single nucleus captured as two or more segmenation volumes), the largest segmentation volume was preserved and the rest deleted. Pyknotic nuclei from dead cells were removed from the data tables and not counted.

Fluorescence intensity decays along the Z-axis, which is as an artefact of the imaging process on a confocal microscope. We modelled this decay for each individual channel and embryo by fitting a linear regression to the logarithm of fluorescence values as a function of the Z-value. However, as the slope of this model is based on a moderate number of cells, it is not stable; therefore, we used an empirical Bayes approach[68] to shrink the coefficients towards the global average of the slopes. Finally, we used these embryo-specific empirical Bayes-corrected slopes to adjust the logarithm values of fluorescence intensity for each nucleus in the embryo (code to perform correction is available, see below). To assess temporal variation in intensity levels across experiments, we plotted the corrected levels of Hoechst, GATA6 and NANOG in ICM cells for each experiment as a function of the experimental date (Supplementary Fig. 2). We observed no temporal trend in fluorescence intensity and a conserved range of fluorescence levels across experiments. We identified one litter where GATA6 and NANOG levels were below the regular range (asterisk in Supplementary Fig. 2b,c), which we excluded from subsequent analysis.

All treatments were repeated in at least two independent litters. Transfer from FGF4 to MEKi (Fig. 5g,h) was done on only one litter. All control embryos were littermates to the corresponding treated embryos on each experiment. Enough animals were used so that each experimental group had a minimum of three embryos (see figures), except for two instances of FGFRi treatment where only two embryos were used (Fig. 2b and 3b), as well as for the transfer from FGF4 to MEKi (Fig. 5g,h). All treatment regime and condition combinations were performed over several months, in no predetermined order.

We tested three approaches to classify cells into lineages. First, for exploratory purposes, all cells were manually assigned to either the TE or one of the ICM lineage denominations (EPI, PrE, DP or DN) when correcting for over- and under-segmentation. Cells were classified as TE based on their location on the periphery of the blastocyst, their morphology, the lack of expression of ICM markers and/or the expression of CDX2, whenever this information was available. ICM cells were visually classified as EPI when showing exclusive expression of NANOG, PrE when showing exclusive expression of GATA6, DP when showing expression of both NANOG and GATA6 (a ratio of GATA6:NANOG approximately between 0.5 and 2) and DN when showing expression of neither marker. However, we found this approach to be inconsistent and subject to investigator bias. We therefore applied a thresholding approach to freshly collected littermates as a reference sample (Fig. 1b). Using this method, we established thresholds of 50 a.u. (arbitrary units) for each NANOG and GATA6 expression (on linear scale, which ranged from 0 to 255 a.u. on an 8-bit image), based on the data distribution in embryos >120 cells, where EPI and PrE populations are completely segregated and cells committed to their respective lineages[15,35]. These thresholds determined four quadrants (EPI: NANOG > 50, GATA6 < 50; PrE: NANOG < 50, GATA6 > 50; DP: NANOG and GATA6 > 50, and DN: NANOG and GATA6 < 50; Fig. 1c). Although this approach is unbiased, it requires the manual determination of thresholds for different imaging parameters and experimental setups, needs access to a comparable reference sample of embryos at all blastocyst stages and relies on linear-scale fluorescence intensity levels, which show a high spread.

To establish an unbiased and more flexible method to assign identities, we devised a data-driven clustering approach that can be directly adapted to data obtained in different systems. When plotting the distribution of ICM cells based on the logarithm of GATA6 and NANOG levels, we observed a clear progression from a single cluster at early blastocyst stages (32–64 cells)—when DP cells dominate the ICM—towards two isolated clusters of NANOG + and GATA6 + cells at late blastocyst stages (120–150 cells)—when both EPI and PrE are completely

segregated (Fig. 1d). We therefore combined the distribution of cells at the 32–64-cell and the 120–150-cell stages, to determine the centre of three clusters—EPI, PrE and DP—using $k$-means clustering. From the centres for the EPI and PrE clusters, we generated the location of the cluster centre for a DN cluster. We then applied these centres to classify cells across all stages such that each cell is assigned to the class of the centre closest to it. Reassuringly, the identities determined using the $k$-means clustering approach essentially recapitulated those obtained using the thresholding approach for the reference littermates (Fig. 1b). We applied these clusters to embryos in Figs 1 and 5b,c and Supplementary Fig. 7a,b, which were acquired with identical parameters. For embryos cultured until the 120–150-cell stage (Figs 2, 4 and 5d,e, and Supplementary Fig. 3), which were acquired with slightly different parameters, and where no or very few DP cells are expected, we used the same method to determine new cluster centres based on the distribution of values for ICM cells of control embryos (Fig. 2b). These values were then applied to classify ICM cells in embryos treated in all conditions. For the subset of embryos shown in Fig. 3, cell identity was only determined manually, as GATA4 and NANOG always show a mutually exclusive expression pattern[15,16], and as GATA4 and GATA6 levels might not necessarily be equivalent in PrE cells.

To estimate the proportion of PrE, DP and EPI + DN cells as a function of the embryo size (total cell number; Supplementary Figs 1e and 6b,c), we used a local regression approach[69].

**Data and code availability.** The authors declare that all data supporting the findings of this study are available within the article and its Supplementary Information, from the authors upon reasonable request and readily available from the following repositories. Corrected data tables for each embryo are freely available for download from Figshare with the identifier http://dx.doi.org/10.6084/m9.figshare.c.3447537.v1 (ref. 70), along with their corresponding original, uncorrected data tables, all raw images for each embryo (in Zeiss' *.lsm format) and their corresponding segmentation files generated by MINS (as *.tiff sequences).

All data analysis was performed on RStudio, as a developing environment for R (version 3.2.2). Tables of the compiled data, experimental references and annotated scripts containing the code used for all data transformations, Z-correction and identity assignment, as well as for generating all plots in the figures, are available at http://github.com/nestorsaiz/saiz-et-al_2016.

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

## Acknowledgements

We thank Minjung Kang and Nadine Schrode for feedback on data processing and presentation and Berenika Plusa, Alberto Puliafito, Stefano DiTalia, Alfonso Martínez Arias, Pau Rué, Christian Schröeter and especially members of the Hadjantonakis lab for comments on the manuscript and data analysis. N.S. is supported by a fellowship from the Tri-Institutional Stem Cell Initiative, funded by The Starr Foundation; K.M.W. is supported by the Louis V Gerstner-Sloan Kettering (GSK) Summer Undergraduate Research Program (SURP). V.E.S. is funded by a Cancer Center support grant (P30 CA008748). Work in the Hadjantonakis lab is funded by NYSTEM (N13G-236) and the by the National Institutes of Health (R01-HD052115 and R01-DK084391).

## Author contributions

N.S. and A.-K.H. conceived and planned the experiments. N.S. and K.M.W. performed the analysis of lineage allocation on freshly collected, fixed embryos. N.S. and V.E.S. performed data analysis and transformations. N.S. performed FGF modulation and ICM scaling experiments, generated the figures and wrote the manuscript. N.S. and A.-K.H. edited the manuscript with input and approval from all authors.

## Additional information

**Competing financial interests:** The authors declare no competing financial interests.

