## [Peer Review File · Nature Communications]

Reviewer #2 (Remarks to the Author)

Reviewer's comments for Saiz et al., "Asynchronous fate decisions by individual cells collectively ensure lineage specification in the mouse blastocyst".

Here, Saiz and colleagues employ confocal image analysis in concert with timed modulation of FGF-MAPK signaling to investigate lineage commitment in the mouse blastocyst. The authors leverage their previously reported quantitative image analysis platform to automate monitoring of GATA6 and NANOG proteins in individual blastomeres of early mouse embryos in which the key regulator of the cell fate choice between the epiblast and primitive endoderm (PrE) in the inner cell mass (ICM), FGF/MAPK signaling, is modulated in a temporal manner. They provide evidence that at least three dynamic cell populations exist in the ICM, including cells co-expressing GATA6 and NANOG. The authors propose, these so-called double-positive cells, represent bipotential progenitors capable of acquiring either epiblast or PrE fate. The central experimental observation is that the ICM composition in cells expressing solely GATA6 (prospective PrE) or NANOG (prospective epiblast) remains relatively constant in the ICM regardless of the ICM number, while the representation of the double positive putative bipotential progenitors decreases over time. Based on elegant temporal manipulations of FGF/MAPK signaling the authors propose that individual cells commit to either epiblast or primitive endoderm fate asynchronously but in coordinated manner. Moreover they posit that a 'seed' of pluripotent epiblast may be necessary to trigger differentiation. Finally, they interpret these observations to suggest that a tissue size control mechanism operates to control lineage size.

Whereas the observation of asynchronous allocation to epiblast or PE cell fate has been previously proposed (Bessonard et al., 2014; Gerbe et al., 2008; Plusa et al., 2008; Laval et al., 2012), as that the requirement for a 'seed' of pluripotent epiblast may be necessary to trigger differentiation (Bessonard et al., 2014; Morris et al., 2013), the combination of automated image analysis with manipulating and analyzing embryos from the same litters, and timed excess or inhibition of FGF/MAPK signaling afford a comprehensive analysis of cell fate decisions within the ICM. The most novel is the concept of asynchronous and incremental but coordinated allocation to the PrE and epiblast lineages through a tissue size control mechanism. This would ensure balanced lineage allocation within the ICM and could provide one mechanism for the famously regulative basis of early mammalian development. The work is carefully done and manuscript clearly presented. Whereas the work does not directly test this model, its overall experimental and conceptual advance will be of interest to a broad scientific community and warrants publication in Nature Communications when the following important issues are addressed.

Major points:

1. The key experimental observation in support of the authors' model is "that the ICM composition in the late blastocyst is highly consistent". In particular "Even though the ICM size was observed to decrease with long culture times (Fig. 3Sc,d), the average ratio of PrE to epiblast cells was kept at or near 3:2 irrespective of culture time (~60% PrE to ~40% epiblast cells (Fig. 2d; S4). These conclusions are based on the automatic analyses of NANOG and GATA6 expression patterns in single cells and the comparison of cultured blastocysts (Fig. 2) to the blastocysts at equivalent developmental stages obtained from females (Fig. 1). This comparison is not easy as different scale is used in Figures 1c and Figure 2b, and equivalent scales should be used.
2. Examination of NANOG and GATA6 whole mount staining in Figures 1b and 2c, shows in some cultured blastocyst (8 cells + 48h; 64 cells + 90 cells) blastomeres expressing NANOG at much higher levels than in uncultured blastocysts containing 120-150 cells. Does this simply reflect a developmental delay (are these blastocysts more similar to 90-120 stage embryos)? How were such cells classified? Can these cells be still classified as epiblast or DP cells? This classification is the foundation of the entire manuscript and the authors acknowledge the limitations of this approach.
3. The figures are excellent, but still difficult to follow and to compare various aspects of cultured, treated and control embryos. In particular, it is not easy to compare the numbers and proportions

of epiblast, PrE, DP and DN cells over time, between cultured and control embryos fixed upon collection. A figure showing the proportions of all blastocyst cell types: TE, Epi, PrE, DP would be informative.

4. One striking observation that is not emphasized by the authors is the difference in the proportion of TE to ICM in cultured blastocysts versus those fixed upon collection. Whereas in embryos fixed upon collection, the TE/ICM ratio is highly constant 60%/40% (Fig. 1f,g and S1), in embryos cultured from 8 cells + 48h, ICM appears to constitute only 20% of the blastocysts, and this difference diminishes for embryos collected at later stages and cultured for shorter periods (Fig. S3c,d). Interestingly, the total number of cells in blastocysts cultured for various time periods appears to be ca. 100-120, and is reduced compared to uncultured blastocyst ~ 2 fold (Figures S1 and S3d). Hence, it appears that ICM growth and/or cell allocation to the ICM is particularly affected.

5. One therefore wonders, whether the consistent ratio of Epiblast/PrE cells in the ICM of these cultured embryos reflects the tissue size control mechanism invoked by the authors. Alternatively, is this a reflection of inhibited ICM growth and developmental delay that preserves the E/PrE composition the authors observe in early blastocyst (Fig. 1d)? It would be important to address whether cell proliferation of the ICM cells is affected in blastocysts cultured at different times and for different time periods. In particular, are the number of ICM cells generated in the first and the second wave of internalization affected? Understanding this is essential given the previous report that the cells generated in the second wave express higher levels of Fgfr2 and are biased towards the PrE fate (Morris et al., 2013).

6. Given the concern that the proposed scalability of Epi:PrE could be based on developmental delays from culture conditions additional tests of the constant Epi:PrE would significantly strengthen this conclusion. Chimeras, or half-embryos would more directly test this hypothesis. The authors' claim here is supported by such previous experiments with half-embryos, where epiblast cell number is highly variable, shows that PrE numbers correlate with size of the Epi (Morris et al., 2012).

7. The manuscript would be strengthened by framing the current data in the context of previous relevant studies. For example, Nichols et al., 2009, and Yamanaka et al., 2010, have made elegant observations from the similar application of FGF/MAPK signal manipulations. Specifically, the Nichols paper has shown that inhibition of Erk activity expands the Epiblast compartment, and that the timing of inhibitor application is critical to the experimental outcomes. Furthermore, previous work demonstrating a role for FGF signaling in Epi/PrE allocation (Morris et al., 2013) is completely overlooked.

8. Methodologically, the authors analyze embryos from the same litter in an attempt to normalize the developmental stage at which FGF-MAPK modulation is initiated. Even within the same litter, embryos can be at different stages, and have different numbers of inside cells generated from the first/second wave of asymmetric division. This should be discussed.

9. The number of inside cells at the 16-cell stage can vary considerably, but the number of inside cells generated by the first and second waves of asymmetric division is regulated in order to produce a more constant number of inside cells at the 32-cell stage. It would be interesting to see how later lineage allocation is influenced by this since the authors suggest that epiblast is seeded and it has been shown previously that this early cell allocation can impact patterning mechanisms (and this may explain why Epiblast:PrE ratio does not stabilize until later in development).

Reviewer #3 (Remarks to the Author)

This paper describes changes in populations of the inner cell mass (ICM) of mouse preimplantation embryos during cell fate specification into epiblast and primitive endoderm (PrE). The authors quantitatively examined the expression of Nanog and Gata6 proteins, the markers for epiblast and PrE, respectively, in individual cells of staged embryos. This analysis revealed gradual reduction of Nanog and Gata6 double-positive (DP) cells from the 32-cell to 120-cell stage. In parallel with this, the effects of manipulation of Fgf4-MAPK signaling, which controls epiblast-PrE fates, gradually decreased up to the 120-cell stage. PrE did not induce epiblasts, but the epiblasts induced PrE.

Based on these results, the authors hypothesized that DP cells are dual-fated ICM progenitor cells and that single-positive cells are committed cells insensitive to Fgf4-MAPK signaling. Therefore, a model was proposed in which commitment of individual ICM progenitors to the epiblast or PrE takes place in an asynchronous manner and a seed of epiblast cells is required for differentiation. This quantitative, single-cell level and dual marker analysis of staged embryos revealed the dynamics of cell populations in the cell-fate-specification process in vivo for the first time. The results are novel and informative, and the overall process described in this paper seems to be correct. However, the current, highly simplified definitions of ICM progenitor cells and committed cells did not align with experimental results. Therefore, the authors' model is not clearly supported by the data.

Comments:

1. Throughout the paper, the authors proposed that the DP cells are ICM progenitors that could respond to Fgf-MAPK signaling and give rise to either epiblast or PrE. The authors also assumed the Nanog or Gata6 single-positive cells to be committed epiblast or PrE cells that are insensitive to Fgf-MAPK signaling. If these hypotheses are true, then the fraction of the cells, which alter cell fates by manipulation of Fgf-MAPK signaling, should be limited to the DP cell population, which are shown in purple in Figure 1e. However, as shown in Figure 2d, clearly, a larger fraction of the cells altered their fates after manipulation of Fgf-MAPK signaling. Therefore, the number of ICM progenitors should be larger than the number of DP cells defined in this paper. Alternatively, not all single-positive cells are committed cells.
2. Related to comment 1: The definition of DP cells seems to be inappropriate. Can all the cells assigned as single-positive cells by the program truly be characterized as single-positive through visual inspection of the images? The graph shows that MANY of the single-positive cells (especially 32-90 cells, Figure 1c) express both markers at higher levels than some DP cells. The same criteria or threshold should be used for both single-positive and double-positive cells to judge for expression of the marker genes.
3. Also related to comment 1: The experiments shown in Figure 4f demonstrated that a large fraction of the Gata6 or Nanog single-positive cells altered their fates from PrE to epiblast and vice versa, respectively, after removal of Fgf ligands or ERK inhibitors. Therefore, it is not appropriate to consider single-positive cells as committed cells. Single-positive cells should contain uncommitted cells or ICM progenitors.
4. Related to comment 3: The authors also noted that single-positive cells can give rise to different cell types under certain conditions. Based on this result, the authors discussed that ICM progenitors may be only a subpopulation of the DP cells defined in this study. However, if ICM progenitors are only a subpopulation of DPs, then the observed fate change of the single-positive cells could not be explained.

Minor comments:

1. Figure 1b does not show representative embryos. In the text, it is described that DP cells are present up to the 120-cell stage. Although this description is consistent with the graphs of Figure 1c, the photographs shown in Figure 1b do not show DP cells except for one cell of the 32- to 64-cell stage embryo. Furthermore, DP cells in photographs should be marked by arrowheads or asterisks to help interpretation of the data.
2. Page 9, the second paragraph. The text says "We cultured all embryos until they reached 120-150 cells, equivalent to the peri-implantation stage (~E4.5)", but in Figure 2a, E4.5 corresponds to "120-170 cells". Which is correct?
3. Page 11, the first paragraph. "Fig. S3c, S5b" may be "Fig. 3c, S5b". Please check.
4. Page 13, the first paragraph. "Fig. 1a, b" may be "Fig. 5a, b". Please check.

Reviewer #2 (Remarks to the Author):

Here, Saiz and colleagues employ confocal image analysis in concert with timed modulation of FGF-MAPK signaling to investigate lineage commitment in the mouse blastocyst. The authors leverage their previously reported quantitative image analysis platform to automate monitoring of GATA6 and NANOG proteins in individual blastomeres of early mouse embryos in which the key regulator of the cell fate choice between the epiblast and primitive endoderm (PrE) in the inner cell mass (ICM), FGF/MAPK signaling, is modulated in a temporal manner. They provide evidence that at least three dynamic cell populations exist in the ICM, including cells co-expressing GATA6 and NANOG. The authors propose, these so-called double-positive cells, represent bipotential progenitors capable of acquiring either epiblast or PrE fate. The central experimental observation is that the ICM composition in cells expressing solely GATA6 (prospective PrE) or NANOG (prospective epiblast) remains relatively constant in the ICM regardless of the ICM number, while the representation of the double positive putative bipotential progenitors decreases over time. Based on elegant temporal manipulations of FGF/MAPK signaling the authors propose that individual cells commit to either epiblast or primitive endoderm fate asynchronously but in coordinated manner. Moreover they posit that a 'seed' of pluripotent epiblast may be necessary to trigger differentiation. Finally, they interpret these observations to suggest that a tissue size control mechanism operates to control lineage size.

Whereas the observation of asynchronous allocation to epiblast or PE cell fate has been previously proposed (Bessonard et al., 2014; Gerbe et al., 2008; Plusa et al., 2008; Laval et al., 2012), as that the requirement for a 'seed' of pluripotent epiblast may be necessary to trigger differentiation (Bessonard et al., 2014; Morris et al., 2013), the combination of automated image analysis with manipulating and analyzing embryos from the same litters, and timed excess or inhibition of FGF/MAPK signaling afford a comprehensive analysis of cell fate decisions within the ICM. The most novel is the concept of asynchronous and incremental but coordinated allocation to the PrE and epiblast lineages through a tissue size control mechanism. This would ensure balanced lineage allocation within the ICM and could provide one mechanism for the famously regulative basis of early mammalian development. The work is carefully done and manuscript clearly presented. Whereas the work does not directly test this model, its overall experimental and conceptual advance will be of interest to a broad scientific community and warrants publication in Nature Communications when the following important issues are addressed.

Major points:

1. The key experimental observation in support of the authors' model is "that the ICM composition in the late blastocyst is highly consistent". In particular "Even though the ICM size was observed to decrease with long culture times (Fig. 3Sc,d), the average ratio of PrE to epiblast cells was kept at or near 3:2 irrespective of culture time (~60% PrE to ~40% epiblast cells (Fig. 2d; S4). These conclusions are based on the automatic analyses of NANOG and GATA6 expression patterns in single cells and the comparison of cultured blastocysts (Fig. 2) to the blastocysts at equivalent developmental stages obtained from females (Fig. 1). This comparison is not easy as different scale is used in Figures 1c and Figure 2b, and equivalent scales should be used.

Indeed, as the reviewer has noted, fluorescence intensities for the datasets in figure 1 and figure 2 are on slightly different scales, as those two cohorts of embryos were acquired with somewhat different parameters (as is now explained in the Methods section of the manuscript). We appreciate the reviewer's concern; however, it should be borne in mind that the scale (i.e., fluorescence intensity range) only affects directly the identity given to each cell (based on marker expression level). Once identity has been assigned (by any given method, as discussed in the manuscript), ICM composition is calculated as the fraction of the number of ICM cells that

each lineage represents. With this issue in mind, our revised, data-driven, identity assignment method, bypasses differences in scale, as the clusters are seeded based on reference embryos for each data set (late-stage blastocysts for Fig. 1 or control, untreated embryos for Fig. 2 – and correspondingly for Figs. 4 and 5, as detailed in the Methods).

2. Examination of NANOG and GATA6 whole mount staining in Figures 1b and 2c, shows in some cultured blastocyst (8 cells + 48h; 64 cells + 90 cells) blastomeres expressing NANOG at much higher levels than in uncultured blastocysts containing 120-150 cells. Does this simply reflect a developmental delay (are these blastocysts more similar to 90-120 stage embryos)? How were such cells classified? Can these cells be still classified as epiblast or DP cells? This classification is the foundation of the entire manuscript and the authors acknowledge the limitations of this approach.

This is a fair concern, as the images shown do, indeed, exhibit cells expressing higher levels of NANOG. However, we believe these differences to be the result of both biological variability between cells and minor differences in acquisition parameters between embryos in Fig. 1 and 2, as mentioned above and explained in the Methods. As also discussed in the Methods, all embryos in each group of experiments were imaged with the same parameters, which means individual cells may exhibit expression levels outside the range of the embryos used to establish the imaging conditions. We believe scatter plots to be a good illustration of variability between embryos, as they show the spread in the data for all embryos – differences in scale notwithstanding, 120-150 cell embryos in Fig. 1d and 2b show an equivalent distribution of data for ICM cells. Furthermore, we have included a new Fig. S2, comparing fluorescence levels between individual experiments. Finally, although developmental delay due to *in vitro* culture is certainly a factor to consider, as we discuss in the main text, we are confident that, as far as these data go, embryos of equivalent cell number are largely comparable between experiments.

3. The figures are excellent, but still difficult to follow and to compare various aspects of cultured, treated and control embryos. In particular, it is not easy to compare the numbers and proportions of epiblast, PrE, DP and DN cells over time, between cultured and control embryos fixed upon collection. A figure showing the proportions of all blastocyst cell types: TE, Epi, PrE, DP would be informative.

We appreciate this comment, as we strive to improve the presentation of these data. We have now incorporated the reviewer's feedback, and presented all lineages (TE, EPI, PrE and DP instead of TE vs. ICM) in Figs. 1e, f; 4c, h; S1d and S4c. Furthermore, we have included regression curves in Figs S1e and S6b, c, showing variation in the size of each compartment as a function of embryo size for a more direct comparison. We hope these additions clarify our results.

4. One striking observation that is not emphasized by the authors is the difference in the proportion of TE to ICM in cultured blastocysts versus those fixed upon collection. Whereas in embryos fixed upon collection, the TE/ICM ratio is highly constant 60%/40% (Fig. 1f,g and S1), in embryos cultured from 8 cells + 48h, ICM appears to constitute only 20% of the blastocysts, and this difference diminishes for embryos collected at later stages and cultured for shorter periods (Fig. S3c,d). Interestingly, the total number of cells in blastocysts cultured for various time periods appears to be ca. 100-120, and is reduced compared to uncultured blastocyst ~ 2 fold (Figures S1 and S3d). Hence, it appears that ICM growth and/or cell allocation to the ICM is particularly affected.

The reviewer raises a very good point. We have noted this stark difference in TE/ICM composition between cultured and non-cultured embryos. As mentioned in the discussion, we believe this is largely due to *in vitro* culture, as the differences are reduced the shorter the culture period and exist in both control and FGF4-treated embryos. At this point we can only

ascribe this effect to differences between intra-uterine and *in vitro* culture conditions (both nutrients and environmental factors, such as pH). Although this difference should be definitely considered in future studies using *in vitro* culture, we think in the present case it provides further support to our observations on consistency in ICM composition, since control embryos from all regimes show a comparable ICM composition (figure 2d), despite having ICMs that differ in overall size.

Regarding the total cell number, embryos cultured for long periods do indeed show a lower cell count than those cultured for shorter periods (compare Fig. S4a far left and far right panels), which we believe to be a reflection of the aforementioned developmental delay. We think this is also the reason why cultured control embryos retain a small fraction of DP cells, whereas embryos freshly isolated and fixed at the 120-150 cell stage show no DP cells or a negligible percentage of them. Both of these issues have now been discussed further in the main text of our manuscript.

5. One therefore wonders, whether the consistent ratio of Epiblast/PrE cells in the ICM of these cultured embryos reflects the tissue size control mechanism invoked by the authors. Alternatively, is this a reflection of inhibited ICM growth and developmental delay that preserves the E/PrE composition the authors observe in early blastocyst (Fig. 1d)? It would be important to address whether cell proliferation of the ICM cells is affected in blastocysts cultured at different times and for different time periods. In particular, are the number of ICM cells generated in the first and the second wave of internalization affected? Understanding this is essential given the previous report that the cells generated in the second wave express higher levels of *Fgfr2* and are biased towards the PrE fate (Morris et al., 2013).

We beg to disagree with the reviewer on the first question, since early blastocysts show a much higher fraction of DP ICM cells than cultured embryos do, with the composition in the latter being comparable to that of mid to late blastocysts (90-150 cells). Instead, we believe the variation in ICM size observed between different culture regimes, as well as those engineered for the new experiments now shown in Fig. 4, provide further support for our model that tissue size can scale with ICM size to maintain a consistent ICM composition. These data are shown also in Fig. S6b and c, which can be directly compared to Fig. S1e.

Regarding the second question, given that we begin most of our experiments at the early blastocyst stage (\Rightarrow 32-cell), when both waves of internalization have already taken place, we believe we cannot ascertain anything regarding the origin of the cells in each compartment. Furthermore, in our experiments using blastocysts, both waves took place *in utero*, thus making them inaccessible to live imaging. Assessing the wave of origin of ICM cells would only be possible for embryos cultured from the 8-cell stage. However, given that in those experiments all ICM cells are forced towards either lineage by the culture conditions (FGF4 or MEKi), we fail to see how the wave of origin would affect the fate decision as assessed in our experimental regime.

6. Given the concern that the proposed scalability of Epi:PrE could be based on developmental delays from culture conditions additional tests of the constant Epi:PrE would significantly strengthen this conclusion. Chimeras, or half-embryos would more directly test this hypothesis. The authors' claim here is supported by such previous experiments with half-embryos, where epiblast cell number is highly variable, shows that PrE numbers correlate with size of the Epi (Morris et al., 2012).

We thank the reviewer for this comment. We have now incorporated an entirely new set of experiments addressing this point. We have performed embryo aggregations at the 8-cell stage, as well as generating half embryos (by dissociating and re-aggregating blastomeres at the 8-cell stage) to double or half the size of the ICM, respectively. These are similar to experiments

carried out by Tarkowski (1961), Buehr and McLaren (1974), Lewis and Rossant (1982) and, barring technical differences, Morris et al (2012). The results of these experiments are presented in Fig. 4 and S6, and support the invariance of ICM composition despite variation in ICM size. We believe that the addition of these experiments significantly strengthens this study and our conclusions.

7. The manuscript would be strengthened by framing the current data in the context of previous relevant studies. For example, Nichols et al., 2009, and Yamanaka et al., 2010, have made elegant observations from the similar application of FGF/MAPK signal manipulations. Specifically, the Nichols paper has shown that inhibition of Erk activity expands the Epiblast compartment, and that the timing of inhibitor application is critical to the experimental outcomes. Furthermore, previous work demonstrating a role for FGF signaling in Epi/PrE allocation (Morris et al., 2013) is completely overlooked.

We appreciate the reviewer's comment and we regret not having discussed these central studies further within the original submission. All these studies have now been discussed further and our data is now contextualized within Nichols', and Morris' work.

8. Methodologically, the authors analyze embryos from the same litter in an attempt to normalize the developmental stage at which FGF-MAPK modulation is initiated. Even within the same litter, embryos can be at different stages, and have different numbers of inside cells generated from the first/second wave of asymmetric division. This should be discussed.

The criteria used to process embryos and litters (as well as to define treatment regimes) has been explained in greater detail now in the Methods section of the manuscript. Furthermore, individual embryos (rather than cell number) are shown in Figs. 1e and g.

9. The number of inside cells at the 16-cell stage can vary considerably, but the number of inside cells generated by the first and second waves of asymmetric division is regulated in order to produce a more constant number of inside cells at the 32-cell stage. It would be interesting to see how later lineage allocation is influenced by this since the authors suggest that epiblast is seeded and it has been shown previously that this early cell allocation can impact patterning mechanisms (and this may explain why Epiblast:PrE ratio does not stabilize until later in development).

We appreciate the reviewer's comment. As discussed above, all blastocysts used in this study were collected at the 32-cell stage or later, when both waves of internalization have been completed. Moreover, embryos in Fig. 5, cultured from the 8-cell stage, are transferred to control media at the 40 or 50-cell stage, and therefore, after both waves have taken place as well. Even though these experiments could be live imaged, we believe only a dual labeling of both lineages with spectrally distinct live reporters would allow to conclusively relate the wave of origin with the final contribution of each ICM cell. We fully recognize the importance of these experiments and we are working towards realizing such goal in the future, however we believe these experiments fall outside the scope of the present study.

Reviewer #3 (Remarks to the Author):

This paper describes changes in populations of the inner cell mass (ICM) of mouse preimplantation embryos during cell fate specification into epiblast and primitive endoderm (PrE). The authors quantitatively examined the expression of Nanog and Gata6 proteins, the markers for epiblast and PrE, respectively, in individual cells of staged embryos. This analysis revealed gradual reduction of Nanog and Gata6 double-positive (DP) cells from the 32-cell to 120-cell stage. In parallel with this, the effects of manipulation of Fgf4-MAPK signaling, which controls epiblast-PrE fates, gradually decreased up to the 120-cell stage. PrE did not induce epiblasts, but the epiblasts induced PrE. Based on these results, the authors hypothesized that DP cells are dual-fated ICM progenitor cells and that single-positive cells are committed cells insensitive to Fgf4-MAPK signaling. Therefore, a model was proposed in which commitment of individual ICM progenitors to the epiblast or PrE takes place in an asynchronous manner and a seed of epiblast cells is required for differentiation.

This quantitative, single-cell level and dual marker analysis of staged embryos revealed the dynamics of cell populations in the cell-fate-specification process *in vivo* for the first time. The results are novel and informative, and the overall process described in this paper seems to be correct. However, the current, highly simplified definitions of ICM progenitor cells and committed cells did not align with experimental results. Therefore, the authors' model is not clearly supported by the data.

Comments:

1. Throughout the paper, the authors proposed that the DP cells are ICM progenitors that could respond to Fgf-MAPK signaling and give rise to either epiblast or PrE. The authors also assumed the Nanog or Gata6 single-positive cells to be committed epiblast or PrE cells that are insensitive to Fgf-MAPK signaling. If these hypotheses are true, then the fraction of the cells, which alter cell fates by manipulation of Fgf-MAPK signaling, should be limited to the DP cell population, which are shown in purple in Figure 1e. However, as shown in Figure 2d, clearly, a larger fraction of the cells altered their fates after manipulation of Fgf-MAPK signaling. Therefore, the number of ICM progenitors should be larger than the number of DP cells defined in this paper. Alternatively, not all single-positive cells are committed cells.

We understand the reviewer's concern about this inconsistency, which we agree with. After discussions with our biostatistician collaborator (V. Sheshan) and his colleagues, we have changed the approach used to assign identities in the revision of the manuscript, as discussed above and extensively within the manuscript. We believe our new clustering method allows the determination of cell identities in a much more robust way, and better captures the cell populations present within the ICM. As a result of this change, and as we discuss in the main text, we now observe that the fraction of committed PrE or epiblast cells at each developmental stage shown in Fig. 1 to be a good predictor of the corresponding fraction of PrE or epiblast upon treatment with MEKi or FGF4 in Fig. 2 (compare Fig. 1h and 2d). In other words, upon FGF4 treatment, the fraction of PrE cells present at the end of the treatment period largely corresponds to the fraction of PrE + DP cells at the stage of treatment initiation, and vice-versa for MEKi. We believe these results better support the proposed model. Nonetheless, we do acknowledge that in the absence of a definitive marker for the progenitor population, the size of such population might not necessarily correspond to that of the DP compartment. However, as we discuss in the manuscript, we do not think this potential difference in the precise size of the progenitor pool detracts from the overall biological mechanisms we put forward.

2. Related to comment 1: The definition of DP cells seems to be inappropriate. Can all the cells assigned as single-positive cells by the program truly be characterized as single-positive through visual inspection of the images? The graph shows that MANY of the single-positive cells (especially 32-90 cells, Figure 1c) express both markers at higher levels than some DP cells. The same criteria or threshold should be used for both single-positive and double-positive

cells to judge for expression of the marker genes.

As mentioned above, we have now changed our approach to identity assignment (which is explained in detail both in the Methods and the Results sections of the manuscript) and we believe the new strategy to be both more robust and more easily applicable to different datasets. We did manually score every single cell analyzed in this study (that information will be made available with the raw data upon publication of the manuscript). However, such a manual scoring is prone to investigator bias and can be affected by variability in fluorescence levels between images. We believe both the clustering and the thresholding methods we have applied in the revised version of the manuscript, since they are data driven and work on corrected data (to account for z-associated fluorescence decay and inter-sample variability, as explained in the Methods), yield a more consistent and reproducible result than manual identification – besides being readily scalable for the large cohorts of embryos that our pipeline is designed to work with.

3. Also related to comment 1: The experiments shown in Figure 4f demonstrated that a large fraction of the Gata6 or Nanog single-positive cells altered their fates from PrE to epiblast and vice versa, respectively, after removal of Fgf ligands or ERK inhibitors. Therefore, it is not appropriate to consider single-positive cells as committed cells. Single-positive cells should contain uncommitted cells or ICM progenitors.

We recognize this inconsistency, which we believe stems from the lineage assignment method. Our new clustering-based identity assignment approach affects the identities in this dataset as well. As discussed in the text, we now observe that embryos cultured in MEKi for 30h retain a large fraction of DP cells, which we think provide the source for PrE cells upon release from the inhibitor. Embryos cultured in FGF4, however, present no, or very few DP cells after 30h, and thus cannot regenerate an epiblast. Transfer to MEKi after FGF4 did, however, regenerate an epiblast, leaving the door open to the possibility that some single positive cells are still functionally progenitors at this stage, or alternatively, that they have not fully committed yet and the high dose of inhibitor can force them towards an epiblast fate. Further characterization of these progenitor cells by expression profiling and further experimental manipulations will be necessary to clarify this point.

4. Related to comment 3: The authors also noted that single-positive cells can give rise to different cell types under certain conditions. Based on this result, the authors discussed that ICM progenitors may be only a subpopulation of the DP cells defined in this study. However, if ICM progenitors are only a subpopulation of DPs, then the observed fate change of the single-positive cells could not be explained.

We believe our response to comment 3 addresses this point as well.

Minor comments:

1. Figure 1b does not show representative embryos. In the text, it is described that DP cells are present up to the 120-cell stage. Although this description is consistent with the graphs of Figure 1c, the photographs shown in Figure 1b do not show DP cells except for one cell of the 32- to 64-cell stage embryo. Furthermore, DP cells in photographs should be marked by arrowheads or asterisks to help interpretation of the data.

We now show an embryo with more DP cells at the 64-90 cell stage in Fig. 1b, and we have labelled DP cells with arrowheads at both the 32-64 and 64-90 cell stages, as suggested.

2. Page 9, the second paragraph. The text says "We cultured all embryos until they reached 120-150 cells, equivalent to the peri-implantation stage (~E4.5)", but in Figure 2a, E4.5 corresponds to "120-170 cells". Which is correct?

120-170 cells is indeed the range where our experimental embryos shown in figure 2 fall (see also Fig. S4a), whereas fixed E4.5 embryos in our dataset fall around 150-170 cells (Fig. S1a). We have now corrected the text to be consistent with the figure.

3. Page 11, the first paragraph. "Fig. S3c, S5b" may be "Fig. 3c, S5b". Please check.

This is indeed the case. This error has been corrected, thanks.

4. Page 13, the first paragraph. "Fig. 1a, b" may be "Fig. 5a, b". Please check.

Either figure could be referred to here. We did mean Figure 1, but Figure 5a, b would be valid too.

Reviewer #2 (Remarks to the Author)

In the revised manuscript the authors made a significant effort and largely addressed the reviewers' questions and criticisms. I remain very supportive of publishing this significant and significantly improved study.

One remaining comment for the authors to consider is that the use of the term cellular commitment is not fully justified experimentally:

Lines 426 - 428 "Overall, these data further indicate that ICM cells that commit to a PrE or epiblast fate do not switch fate as a result of changes in FGF4-RTK-MAPK activity".

Lines 534-535 "We show that progenitor cells within the ICM become committed to one or the other lineage (epiblast or PrE) in an asynchronous manner".

Cellular commitment is an operational term that describes the ability of a cell to maintain its predicted fate (by location or gene expression) to develop this fate even when challenged by a different environment e.g. after transplantation. Here, the commitment to either epiblast or PrE fate is inferred from experimental data from fixed embryos and experimental manipulation of FGF signaling. Definitive conclusion of cell fate commitment would require transplanting Nanog⁺/Gata6⁻ or Gata6⁺/Nanog⁻ cell to another embryo and following their individual fates. However, as the authors acknowledge (lines 636-637) "These populations cannot, however, be distinguished using single lineage live imaging reporters". Therefore, it would be important to note the limitations of these experiments and be more cautious with definitive conclusions about Epi and PrE cell fate commitment.

Referee #3 raised a number of questions regarding the model proposed by the authors, about the commitment/developmental potential of both single and double positive (DP) cells in the mouse blastocyst. The Referee noted in particular that a larger fraction of the cells altered their fates after manipulation of Fgf-MAPK signaling than the number of cells identified as DPs by the automatic assignment method employed by the authors. Other questions of this Referee and echoed by my comments questioned the assignment of cells as single or DPs given the large differences in staining intensity presented in the figures.

The authors agreed with these comments and addressed them by changing their approach to assign cell identities. They introduced a new clustering method that "allow the determination of cell identities in a much more robust way, and better captures the cell populations within the ICM". This new approach produces the numbers of DPs and single positive cells better matching the experimental predictions on the FGF pathway inhibition/activation. The authors are also more cautious in their interpretation and discussion of the data by acknowledging that "that in the absence of a definitive marker for the progenitor population, the size of such population might not necessarily correspond to that of the DP compartment".

I therefore think, that the authors have largely sufficiently addressed the valid concerns of Referee #3. One remaining concern is too strong use of the word commitment what Referee #3 points to (comment #3) and what I noted in my comments on the revised ms. While some of this criticism has been addressed by the authors through modification of their cell identity assignment, one is concerned that the limited number of experimental manipulations does not fully test cellular commitment. Indeed, Notch pathway has been also implicated in Epi - PrE decision but has not been studied here. This can be addressed by more cautious discussion.

Response to reviewers

We would like to thank once again the anonymous reviewer for their insightful comments, which we wholeheartedly believe have improved our manuscript and study.

We agree with both reviewers regarding our generous use of the term 'commitment' to describe the state of PrE and epiblast cells in the blastocyst. While we do believe these cells are committed at this stage, as Reviewer #2 has rightfully pointed out, "Cellular commitment is an operational term that describes the ability of a cell to maintain its predicted fate (by location or gene expression) to develop this fate even when challenged by a different environment e.g. after transplantation", which we have certainly not proven experimentally in the present study. We have therefore avoided the use of this term throughout the manuscript, and specifically within the sentences pointed out by the reviewer.

Lines 426-428 now read:

"Overall, these data further indicate that ICM cells that acquire a PrE or epiblast fate do not switch fate as a result of changes in FGF4-RTK-MAPK activity, as they maintain the marker expression pattern of *bona fide* PrE and epiblast cells."

While lines 534-535 now read:

"We show that bipotential progenitor cells within the ICM (DP) become allocated to one or the other lineage (epiblast or PrE) in an asynchronous manner during blastocyst development (Fig. 6a)."

Moreover, we have added the following line to the discussion:

"We therefore propose that *in vivo*, once committed, PrE and epiblast cells can no longer be affected by changes in extracellular FGF4 produced by epiblast cells. In this way, lineage committed cells would become sheltered from differentiation signals, which could otherwise alter their identity – albeit the effect of other signaling pathways, which could potentially alter cell fate at this stage, has not been addressed here and cannot therefore be completely ruled out."

We believe these changes make our text more rigorous and precise and we look forward to experimentally addressing these questions in the future.

The authors.